# Novel fungal metabolites as dual cholinesterase inhibitors: A computational approach for Alzheimer's disease therapy

Md. Habib Ullah Masum[1]*, Syed Mohammad Lokman[2], Rehana Parvin[3], Md Shahidur Rahman[4], Erfanul Haq Chowdhury[5], Kazi Chamonara[6], Salma Chowdhury[7], Ahmad Abdullah Mahdeen[4], Mst. Mitu Khatun[3]

1 Department of Genomics and Bioinformatics, Faculty of Biotechnology and Genetic Engineering, Chattogram Veterinary and Animal Sciences University, Khulshi, Bangladesh, 2 Asian University for Women (AUW), Chattogram, Bangladesh, 3 Department of Pathology and Parasitology, Faculty of Veterinary Medicine, Chattogram Veterinary and Animal Sciences University, Khulshi, Bangladesh, 4 Department of Microbiology, Noakhali Science and Technology University, Noakhali, Bangladesh, 5 Lovely Professional University, Phagwāra, Punjab, India, 6 Department of Environmental Biotechnology, Faculty of Biotechnology and Genetic Engineering, Chattogram Veterinary and Animal Sciences University, Khulshi, Bangladesh, 7 Department of Industrial Biotechnology, Faculty of Biotechnology and Genetic Engineering, Chattogram Veterinary and Animal Sciences University, Khulshi, Bangladesh

* habibullahmasummbg5005@gmail.com, mhumasum@cvasu.ac.bd

**Data availability statement:** All relevant data are within the manuscript and its Supporting Information files.

**Funding:** The author(s) received no specific funding for this work.

## Abstract

Alzheimer's disease (AD), a progressive neurodegenerative disorder, is a major global health concern, affecting millions worldwide, with its prevalence expected to triple by 2050. Despite the widespread use of traditional drugs like cholinesterase inhibitors and NMDA receptor antagonists, their limited effectiveness requires innovative therapeutic approaches. This work used Computer-Aided Drug Design (CADD) to renovate AD therapies aimed at both acetylcholinesterase (AChE) and butyrylcholinesterase (BuChE) using fungal secondary metabolites. Subsequent pharmacokinetic profiles indicated that all metabolites had significant gastrointestinal absorption, blood-brain barrier permeability, and adherence to Lipinski's Rule of Five, suggesting favourable drug-like properties. Furthermore, these metabolites exhibited little toxicity, except for Lovastatin, which indicated possible carcinogenicity. Molecular docking revealed three main candidates—Fumitremorgin C, Hericenone J, and Lovastatin—with notable binding affinities for AChE and BuChE. Consequently, the Fumitremorgin C showed the highest affinity for AChE (−10.0 kcal/mol), but Hericenone J showed enhanced inhibition of BuChE (−9.2 kcal/mol), suggesting its potential use in advanced stages of AD. Molecular dynamics simulations spanning 100 ns validated the stability of enzyme-ligand complexes, with Hericenone J exhibiting the greatest stability, low RMSD, and strong hydrogen bond interactions. The RMSF analysis further demonstrated that Hericenone J preserved structural integrity, whereas ROG and SASA values validated its compactness and stability. As determined by binding

**Competing interests:** The authors declare that they have no known competing financial interests or personal relationships that could have appeared to influence the work reported in this paper.

energy calculations, Hericenone J had the most inhibitory potential, followed by Lovastatin. However, Hericenone J's constant adoption of low-energy conformations, as shown by the principal component and Gibbs free energy analyses, suggested robust and stable interactions with both cholinesterases. With its superior pharmacokinetic profiles, significant binding affinity, and high stability, Hericenone J is the most promising dual cholinesterase inhibitor. These results support the notion that Hericenone J might be an effective treatment for AD if subjected to more preclinical trials.

## 1. Introduction

Alzheimer's disease (AD) is a progressive and irreversible neurodegenerative condition characterized by the degeneration and demise of neurons in the human brain [1]. The predominant kind of AD is dementia, marked by a deterioration in cognitive abilities, deficits in learning and memory, and a reduction in emotional and sensory processes [2,3]. The primary demographics of dementia include age, sex, and having with Down syndrome, with age serving as the most significant risk factor for the onset of AD [1]. AD is a significant global health concern, impacting 12.8% of individuals aged over 65 and 35–40% of individuals over 80, with an estimated 46.8 million people afflicted worldwide, which is projected to treble by 2050 [3,4]. It ranks as the sixth leading cause of death in the United States, with approximately 4% of fatalities attributed to AD [5]. Based on the Alzheimer's Research United Kingdom report, the AD was the major cause of death in 2022 [6]. The precise mechanism of AD remains controversial despite several theories being proposed to explain its underlying cause [3]. The histopathological hallmarks of the disease include the accumulation of extracellular amyloid-beta (Aβ) plaques [3,7], and the formation of intracellular neurofibrillary tangles (NFTs) caused by hyperphosphorylated tau protein, leading to impaired neuronal transport, primarily in the hippocampus and cerebral cortex [7]. Oxidative stress has also been suggested to play a pivotal role in the early stages of AD. These plaques not only trigger oxidative stress and chronic inflammation but also disrupt normal brain function by modulating N-methyl-D-aspartate (NMDA) receptors, causing excitotoxicity and further contributing to neuronal damage [2]. Additionally, cholinergic dysfunction due to reduced acetylcholine (ACh) level [8], mitochondrial dysfunction, progressive loss of synapses and neurons results in brain atrophy, leading to memory impairment, cognitive decline, and death, which occurs in AD [9,10].

Alzheimer's patients survive an average of 7–10 years after showing the symptom [11] and currently have no unequivocal premortem diagnosis, but can only be diagnosed histologically postmortem, which makes it more difficult in early screening and even treatment [12]. Currently, no therapeutic interventions are available to halt or reverse the disease progression [13]. The current pharmacological treatments for AD, including donepezil, rivastigmine, galantamine, and memantine, primarily offer symptomatic relief without altering the underlying disease progression [14]. These agents exert modest yet consistent effects in mitigating reduced cognitive abilities. However, their impact remains limited and associated with diverse side effects [13,15],

highlighting the urgent need for disease-modifying therapies that can prevent neurodegenerative processes [14]. Understanding the molecular mechanisms underlying AD is crucial for developing new therapeutic strategies. Current research on novel therapeutics for AD focuses on targeting various molecular mechanisms and phenomena including amyloid deposition [7], astrogliosis [16], tau protein hyperphosphorylation and accumulation [7], neuronal dystrophy [17], oxidative stress [18], biometal dyshomeostasis, and reduced acetylcholine (ACh) levels [1,2]. Based on the findings from Ach, restoring ACh levels with the combined use of acetylcholinesterase (AChE) and butyrylcholinesterase (BuChE) inhibitors may constitute a successful therapy strategy for AD. Although AChE is the predominant cholinesterase in healthy brains, the activity of BuChE increases in the brains of AD patients. Thus, inhibition of AChE and BuChE should be targeted to effectively enhance the level of ACh in the synaptic cleft, while inhibition of the cholinesterases to elevate the level of ACh is the gold standard symptomatic treatment of the disease [19].

In recent years, medicinal fungi have emerged as a promising source of bioactive secondary metabolites with potential therapeutic applications for many complicated diseases. Several metabolites derived from fungi have demonstrated the ability to protect neurons by influencing signaling pathways, inhibiting AChE and BuChE, and decreasing oxidative damage [1,8]. These compounds are significant as they can hinder Aβ aggregation, an essential step in developing extracellular plaques that lead to neuronal damage, while also managing the pathways related to Aβ production and clearance [20]. The diverse chemical structures of fungal metabolites offer unique scaffolds, and their multi-targeted therapeutics can address the complex pathological mechanisms of AD, making medicinal fungi a promising and effective tool for drug discovery.

Based on the aforementioned properties, the study employed computer-aided drug design (CADD) techniques, including molecular docking and pharmacoinformatic approaches, to identify and screen fungal metabolites derived from fungal medicinal plants as potential therapeutic agents against AD. The study aimed to analyze the interaction patterns of these compounds with key therapeutic targets, providing insights into their potential efficacy in combating the disease.

## 2. Methodology

### 2.1 Retrieval of fungal secondary metabolites, ligand, and receptor protein preparation

In this study, fungal metabolites were identified and retrieved from the Medicinal Fungi Secondary Metabolite And Therapeutics (MeFSAT) (https://cb.imsc.res.in/mefsat/) [21]. From the database, a total of 1830 secondary metabolites were identified from 184 medicinal fungi medicinal; each chosen for possible therapeutic potential against AD. The chemical structures of the fungal metabolites were obtained in PDB format from the server database to prepare the ligand. These structures were further optimized and visualized by PyMOL (https://www.pymol.org/) software. For the receptor protein, the tertiary structures of the AChE (PDB ID: 4EY6) [22,23] and BuChE (PDB ID: 1P0I) [23,24] were obtained from the Protein Data Bank (PDB) (https://www.rcsb.org/structure/5w1j) database. Before molecular docking, the protein structure underwent refinement via GalaxyRefine (https://galaxy.seoklab.org/cgi-bin/submit.cgi?type=REFINE) [25] and was further optimized using AutoDock Tools to enhance docking accuracy (http://vina.scripps.edu/) [26].

### 2.2 Molecular docking analysis

Molecular docking is a fundamental technique in structure-based drug design, commonly used in computer-aided drug design (CADD) to predict optimal binding conformations of micromolecules (ligands) to target macromolecules (receptor protein) [27]. In this study, blind docking was performed to identify potential binding sites on receptor proteins (AChE and BuChE) by using the PyRx (https://pyrx.sourceforge.io/) software [28]. The PyRx integrates AutoDock 4 and AutoDock Vina, enhancing its accessibility and reliability for drug discovery research [26,29]. During docking, the dimensions of the grid box for AChE were defined as X:Y:Z = 56.78:75.17:75.87, with the center also configured at X:Y:Z = 3.82:-62.56:-25.29. For the BuChE, the grid box dimensions were set as X:Y:Z = 68.22:68.23:87.06, with the center also defined as

X:Y:Z = 135.35:124.37:36.34. Donepezil, rivastigmine, and galantamine were employed as standard inhibitors for AChE and BChE, utilizing the grid generation protocols previously outlined [30]. The subsequent docking analysis proceeded with an exhaustive value of 24 [31]. The receptor-ligand interacting residues were further visualized and analyzed with BIOVIA Discovery Studio Visualizer v19.1.0.18287 [32,33].

## 2.3 Pharmacokinetic (PK) profile and toxicity analysis

The pharmacokinetic (PK) profiles of the fungal secondary metabolites, including absorption, distribution, metabolism, and excretion (ADME) were evaluated using SwissADME http://www.swissadme.ch/index.php) server. Following molecular docking, the SMILES notations were retrieved from the PubChem database (https://pubchem.ncbi.nlm.nih.gov/) and subjected to PK assessment using the SwissADME (http://www.swissadme.ch/index.php) [34]. The server predicts key descriptors, including lipophilicity, solubility, gastrointestinal absorption, blood-brain barrier permeability, drug-likeness, and Lipinski's Rule of Five. Additionally, it evaluates interactions with cytochrome P450 enzymes and identifies potential issues such as low solubility or bioavailability [35].

Early toxicity prediction proves crucial for optimizing lead compounds and reducing failure risks in drug development [36]. The potential toxicity of the selected fungal metabolites was evaluated by using the ProTox 3.0 server (https://tox.charite.de/protox3/), which predicts the toxicological profiles of bio-compounds using computational algorithms and extensive toxicological datasets [37,38]. It provides predictions for various toxicity endpoints, including oral toxicity, hepatotoxicity, and carcinogenicity, and categorizes compounds into six toxicity classifications [37,38].

## 2.4. Structure-activity relationship (QSAR) analysis

The quantitative structure-activity relationship (QSAR) analysis of the fungal secondary metabolites was performed by the ChemDes (http://www.scbdd.com/chemdes/) server [39]. The server generates a diverse array of molecular descriptors and fingerprints essential for QSAR analysis, encompassing physicochemical, topological, geometrical, and electronic characteristics to facilitate the correlation between chemical structure and biological activity [40].

## 2.5 Molecular dynamic simulation

Regarding the receptor protein-ligand complexes, the GROningen MAchine for Chemical Simulations (GROMACS) version 2022.3 was employed to perform molecular dynamics (MD) simulations [41,42]. During the simulation run, the ff19SB force field was selected due to its effectiveness in accurately depicting the molecular interactions within the system [43]. Subsequently, a cubic water box was built utilizing the OPC water model to simulate a virtual physiological environment [44,45]. Sodium ($Na^+$) and chloride ($Cl^-$) ions were introduced to equilibrate the net charge of the system. Following this, energy minimization was performed using two consecutive methods: 500 iterations of the steepest descent followed by 1000 iterations of the conjugate gradient approach. The system further achieved a thermal equilibrium state by heating at 300 K via a sequence of isothermal-isochoric (NVT) and isobaric (NPT) equilibration phases, maintaining a pressure of 1.0 bar. The simulation was conducted with a time integration phase of 2 femtoseconds (fs) and periodic boundary conditions. Long-range interactions were stabilized using the particle-mesh Ewald (PME) approach; SHAKE constraints were used to calculate hydrogen bond lengths, maximizing computational efficiency accordingly [46,47]. Proximity interactions were explained using a non-bonded interaction cutoff radius of 10 Å, thus significantly improving system stability. Finally, the simulation was performed for 100 nanoseconds (ns).

## 2.6 Post-simulation analysis

Following the successful simulation run, post-simulation analyses, including root mean square deviation (RMSD), root mean square fluctuation (RMSF), radius of gyration (ROG), solvent-accessible surface area (SASA), and hydrogen bond assessments, were conducted using the relevant modules integrated within the GROMACS software. To further explore

the binding interactions within the docked complexes, molecular mechanics energies integrated with Poisson-Boltzmann surface area (MMPBSA) continuum solvation simulations were performed using the MMPBSA.py program [48]. This approach was utilized to measure molecular interaction energies, and binding free energies on 1 ns trajectory provided a robust estimate of binding dynamics across time [49]. Further, the receptor and its complexes were visualized using PyMOL [50], and VMD [51] software. The XMGRACE software was utilized for plotting graphs, while the Gibbs free energy plots were generated using the Python-based Matplotlib package [52,53].

## 3. Results

The overview of the study is depicted in **Fig 1**.

### 3.1 Molecular docking analysis

A total of 1830 fungal secondary metabolites were retrieved from the MeFSAT database, with their tertiary structures downloaded in PDB format and prepared as ligands. Before molecular docking, the AChE and BuChE protein structure was refined by removing water molecules, metal ions, and cofactors, adding polar and nonpolar hydrogens, and calculating Gasteiger charges. The final protein structures were formatted as PDBQT with AutoDock Tools, and docking analyses were performed using PyRx. A blind docking approach was employed to identify potential binding sites of the receptor protein AChE and BuChE subsequently. The binding affinities of all screened fungal secondary metabolites ranged between −6.0 and −10.0 kcal/mol. Based on their docking scores, six fungal metabolites including Fumitremorgin C, Hericenone J, Lovastatin, Erinacerin M, N-de(phenylethyl) isohericerin, and Hericenone A from *Hericium erinaceus* with binding affinities ≤ −8.2 kcal/mol were considered to have strong interactions and were selected for further virtual screening and analyses (S1 Table).

### 3.2 Pharmacokinetic (PK) profile and toxicity analysis

To further refine the PK profiles of six fungal metabolites; Fumitremorgin C, Hericenone J, Lovastatin, Erinacerin M, N-de(phenylethyl) isohericerin, and Hericenone A, assessments were conducted using the SwissADME server. Based on the SwissADME results, the PK profiles, including lipophilicity (Log Po/w) of the selected fungal metabolites ranging from 1.12 to 4.23, revealed varying degrees of lipophilicity among the compounds. Notably, fungal metabolites such as Hericenone J, Lovastatin, Erinacerin M, and N-de(phenylethyl) isohericerin, exhibited enhanced membrane permeability, suggested by their Log Po/w values (3–5). The metabolites displayed poor water solubility, with Log S (ESOL) values ranging from −4.94 to −2.16; among them, Hericenone A (−2.16) demonstrated the highest solubility. All six compounds exhibited high gastrointestinal (GI) absorption and blood-brain barrier (BBB) permeability, while none were identified as P-glycoprotein (P-gp) substrates. The synthetic accessibility scores ranged from 3.15 to 4.16, indicating moderate synthetic complexity, with Hericenone A (3.15) being the most accessible. Additionally, all metabolites met lead-likeness criteria (S2 Table), fully complied with Lipinski's Rule of Five with no violations, and satisfied the drug-likeness and pharmacokinetic feasibility criteria established by Ghose, Veber, Egan, and Muegge. Furthermore, each compound demonstrated moderate oral absorption potential, with a bioavailability score of 0.55 (S3 Table).

The toxicity assessment of the fungal metabolites was conducted through the ProTox 3.0 web server to evaluate their drug safety profiles. The results indicated that all the compounds were lacking hepatotoxicity, mutagenicity, cytotoxicity, and carcinogenicity, except for Lovastatin, which showed minimal carcinogenicity (S4 Table).

### 3.3 Structure-activity relationship (QSAR) analysis

The QSAR evaluation of six fungal metabolites revealed crucial molecular descriptors that provide insights into the compounds' structural features influencing bioactivity. The Chiv5 index ranged from 1.40 to 3.82, and the bcutm1 values spanned from 3.90 to 4.12. Descriptors MRVSA9 and MRVSA6 values range from 5.91 to 22.72 and 23.80 to 46.06, respectively, and the electrostatic potentials indicator PEOEVSA5 range from 11.65 to 45.92. The GATSv4 values and J

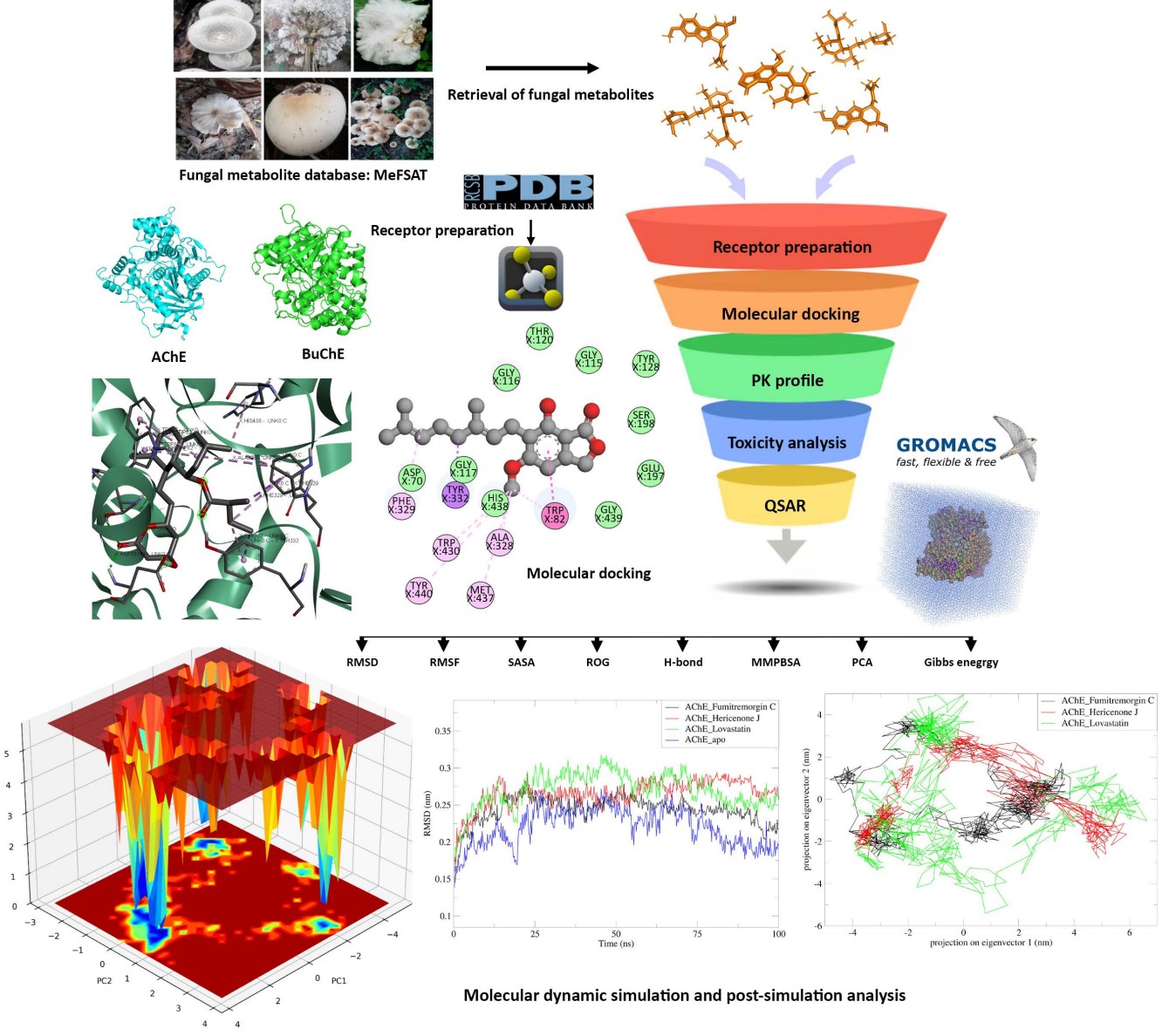

**Fig 1. The overview of the entire study.**

index ranged from 0.98 to 1.24 and 1.46 to 2.15, respectively, and Diametert varies from 7.00 to 13.00. The pIC50 values ranged from 4.41 to 5.21, with Lovastatin demonstrating the highest potency, making it a particularly promising lead compound (**Table 1**). Based on the docking scores, PK profiles, and QSAR analysis, three fungal metabolites, including Fumitremorgin C, Hericenone J, and Lovastatin, were considered for subsequent analysis.

## 3.4 Post-docking analysis

The post-docking analysis of fungal metabolites Fumitremorgin C, Hericenone J, and Lovastatin with AChE and BuChE revealed strong binding affinities. The docking scores for AChE_Fumitremorgin C and BuChE_Fumitremorgin C were −10.0 kcal/mol and −9.1 kcal/mol, respectively, The AChE_Hericenone J and BuChE_Hericenone J showed scores of

**Table 1. The QSAR analysis of the selected fungal metabolites by ChemDes server.**

| Fungal metabolite | Chiv5 | bcutm1 | MRVSA9 | MRVSA6 | PEOEVSA5 | GATSv4 | J | Diametert | pIC50 |
|---|---|---|---|---|---|---|---|---|---|
| Fumitremorgin C | 3.82 | 4.12 | 22.72 | 41.11 | 11.65 | 1.24 | 1.46 | 12.00 | 4.86 |
| Hericenone J | 1.64 | 3.97 | 5.97 | 46.06 | 23.30 | 1.04 | 1.84 | 12.00 | 4.89 |
| N-de(phenylethyl)isohericerin | 1.70 | 3.97 | 5.91 | 46.06 | 23.30 | 1.12 | 1.83 | 13.00 | 4.97 |
| Lovastatin | 3.80 | 3.90 | 11.94 | 23.80 | 45.92 | 0.98 | 1.68 | 13.00 | 5.21 |
| Hericenone A | 1.55 | 3.97 | 11.75 | 46.06 | 17.22 | 1.16 | 1.86 | 13.00 | 4.77 |
| Erinacerin M | 1.40 | 4.01 | 11.75 | 33.74 | 13.85 | 1.07 | 2.15 | 7.00 | 4.41 |

−8.0 kcal/mol and −9.2 kcal/mol, respectively, while AChE_Lovastatin and BuChE_Lovastatin exhibited scores of −9.2 kcal/mol and −8.6 kcal/mol, respectively. On the other hand, the control drugs—galantamine, rivastigmine, and donepezil—demonstrated docking scores of −8.1 kcal/mol, −6.9 kcal/mol, and −9.2 kcal/mol for AChE, and −9.2 kcal/mol, −6.8 kcal/mol, and −10.1 kcal/mol for BuChE, respectively (Table 2). Further intermolecular interaction analysis highlighted multiple molecular interactions in each complex (Table 2, Fig 2).

In the AChE_Fumitremorgin C complex demonstrated Van der Waals interactions with GLN-290, PHE-294, PHE-337, SER-292, TRP-341, TYR-123, TYR-336, and TYR-340. Furthermore, conventional hydrogen bonds were formed with TYR-123 and SER-292, and no carbon-hydrogen bonds were detected. Similarly, the BuChE_Fumitremorgin C complex, the ligand exhibited numerous Van der Waals interactions with ASN-68, ASP-70, GLN-67, GLY-117, HIS-438, ILE-69, LEU-286, PHE-329, PHE-398, PRO-84, SER-198, THR-120, TRP-82, TRP-231, and TYR-332. Additionally, conventional hydrogen bonds were observed with ASP-70 and HIS-438, while carbon-hydrogen bonding occurred with ASN-68 and GLN-67 (Table 2, Fig 2). The AChE_Hericenone J complex exhibited Van der Waals interactions with GLY-291, PHE-294, PHE-337, SER-292, TRP-285, TYR-123, TYR-336, and TYR-340. Additionally, carbon-hydrogen bonds were identified with TYR-123 and SER-292, while a conventional hydrogen bond was formed with GLY-290. In contrast, the BuChE_Hericenone J complex displayed substantial Van der Waals interactions with ALA-328, ASP-70, GLY-117, HIS-438, MET-437, PHE-329, TRP-82, TRP-430, and TYR-440, with no carbon-hydrogen or conventional hydrogen bonds observed (Table 2, Fig 2). The AChE_Lovastatin complex displayed Van der Waals interactions with ARG-295, GLN-290, PHE-337, TRP-285, TYR-123, TYR-340, and VAL-293. Carbon-hydrogen bonds were formed with GLN-290 and TYR-123; no conventional hydrogen bonds were observed. Conversely, the BuChE_Lovastatin complex demonstrated Van der Waals interactions with ALA-328, ASP-70, HIS-438, ILE-69, PHE-329, THR-120, TRP-82, and TYR-332. Carbon-hydrogen bonds were observed with ASP-70 and THR-120, while no conventional hydrogen bonds were detected (Table 2, Fig 2). The AChE_Galantamine complex has key interactions involving residues TYR-71, ASP-73, TRP-285, LEU-288, TYR-340, TYR-123, SER-292, VAL-293, PHE-294, ARG-295, PHE-296, PHE-337, with carbon-hydrogen bonds established between ASP-73 and TYR-340. The BChE_Galantamine complex established conventional hydrogen bonds with SER-198 and HIS-438, interacting with several active site residues, including TRP-82, GLY-115, GLY-116, GLY-117, THR-120, TYR-128, GLU-197, SER-198, ALA-199, TRP-231, TYR-332, PHE-398, PHE-329, HIS-438, and GLY-439 (Table 2). The AChE_Rivastigmine complex was stabilized by conventional hydrogen bonds between PHE-294 and ARG-295 and carbon-hydrogen bonds between ASP-73 and TYR-340. In BChE, rivastigmine engaged with residues including GLY-116, GLY-117, GLN-119, SER-198, TRP-231, SER-287, VAL-288, LEU-286, PHE-329, PHE-398, and HIS-438, establishing a conventional hydrogen bond with GLN-119 and a carbon-hydrogen bond with GLY-117 (Table 2). The donepezil formed several interactions with essential catalytic site residues in both enzymes. In AChE, conventional hydrogen bonds were identified with ASP-73, TYR-123, TRP-285, LEU-288, GLU-291, GLN-290, SER-292, VAL-293, ARG-295, PHE-294, PHE-296, TYR-336, PHE-337, and TYR-340, but a carbon-hydrogen bond was established with ASP-73. In BChE, donepezil interacted with many residues, notably ASP-70, TRP-82, GLY-116, GLY-117, GLN-119, THR-120, GLU-197, SER-198, TRP-231,

**Table 2. The docking scores and intermolecular interaction of the AChE-ligand, and BuChE-ligand complexes.**

| Docked complex | Docking score | Interaction residues | Conventional H-bonds | Carbon H-bonds |
|---|---|---|---|---|
| AChE_Fumitremorgin C | −10.0 | GLN-290, PHE-294, PHE-337, SER-292, TRP-341, TYR-123, TYR-336, TYR-340 | TYR-123, SER-292 | – |
| BuChE_Fumitremorgin C | −9.1 | ASN-68, ASP-70, GLN-67, GLY-117, HIS-438, ILE-69, LEU-286, PHE-329, PHE-398, PRO-84, SER-198, THR-120, TRP-82, TRP-231, TYR-332 | ASP-70, HIS-438 | ASN-68, GLN-67 |
| AChE_Hericenone J | −8.2 | GLY-291, PHE-294, PHE-337, SER-292, TRP-285, TYR-123, TYR-336, TYR-340 | TYR-123, SER-292 | GLY-290 |
| BuChE_Hericenone J | −9.0 | ALA-328, ASP-70, GLY-117, HIS-438, MET-437, PHE-329, TRP-82, TRP-430, TYR-440 | – | – |
| AChE_Lovastatin | −9.2 | ARG-295, GLN-290, PHE-337, TRP-285, TYR-123, TYR-340, VAL-293 | GLN-290, TYR-123 | – |
| BuChE_Lovastatin | −8.6 | ALA-328, ASP-70, HIS-438, ILE-69, PHE-329, THR-120, TRP-82, TYR-332 | ASP-70, THR-120 | – |
| AChE_Galantamine | −8.1 | TYR-71, ASP-73, TRP-285, LEU-288, TYR-340, TYR-123, SER-292, VAL-293, PHE-294, ARG-295, PHE-296, PHE-337 | – | ASP-73, TYR-340 |
| BuChE_Galantamine | −9.2 | TRP-82, GLY-115, GLY-116, GLY-117, THR-120, TYR-128, GLU-197, SER-198, ALA-199, TRP-231, TYR-332, PHE-398, PHE-329, HIS-438, GLY-439 | SER-198, HIS-438 | – |
| AChE_Rivastigmine | −6.9 | TYR-71, ASP-73, TYR-123, TRP-285, LEU-288, SER-292, VAL-293, ARG-295, PHE-294, PHE-296, TYR-336, PHE-337, TYR-340 | PHE-294, ARG-295 | ASP-73, TYR-340 |
| BuChE_Rivastigmine | −6.8 | GLY-116, GLY-117, GLN-119, SER-198, TRP-231, SER-287, VAL-288, LEU-286, PHE-329, PHE-398, HIS-438 | GLN-119 | GLY-117 |
| AChE_Donepezil | −9.2 | ASP-73, TYR-123, TRP-285, LEU-288, GLU-291, GLN-290, SER-292, VAL-293, ARG-295, PHE-294, PHE-296, TYR-336, PHE-337, TYR-340 | TYR-123, PHE-294, ARG-295 | ASP-73 |
| BuChE_Donepezil | −10.1 | ASP-70, TRP-82, GLY-116, GLY-117, GLN-119, THR-120, GLU-197, SER-198, TRP-231, VAL-288, LEU-286, PHE-329, PHE-398, HIS-438, TYR-332, MET-437, ALA-328, TYR-438, TYR-440, TRP-430, GLY-439 | – | GLU-197 |

VAL-288, LEU-286, PHE-329, PHE-398, HIS-438, TYR-332, MET-437, ALA-328, TYR-438, TYR-440, TRP-430, and GLY-439, establishing a carbon-hydrogen link with GLU-197 but lacking typical hydrogen bonds (Table 2).

### 3.5 Molecular dynamic simulation and post-simulation analysis

**3.5.1. RMSD.** The AChE_Fumitremorgin C, AChE_Hericenone J, AChE_Lovastatin, BuChE_Fumitremorgin C, BuChE_Hericenone J, and BuChE_Lovastatin, exhibit significant trends in their complexes throughout a 100 ns simulation period. The RMSD measures the extent of divergence of a group of atoms from the exact reference structure of a protein, ligand, or ligand-protein complex. Elevated RMSD values may indicate a significant level of instability due to changes in the structure of the examined molecule. The average RMSD values for AChE_Fumitremorgin C, AChE_Hericenone J, AChE_Lovastatin, BuChE_Fumitremorgin C, BuChE_Hericenone J, and BuChE_Lovastatin, were found to be $0.24 \pm 0.19$ nm, $0.27 \pm 0.19$ nm, $0.27 \pm 0.02$ nm, $0.25 \pm 0.19$ nm, $0.26 \pm 0.01$ nm, and $0.24 \pm 0.18$ nm, respectively (Fig 3, Table 3).

The fungal metabolites exhibited notable differences in RMSD for the AChE. For the AChE_Fumitremorgin C, the RMSD exhibited a pronounced increase during the first 23 ns (0.29 nm), achieving a steady plateau with few fluctuations until the 52nd ns. At 53 ns, a transitory rise to 0.27 nm was seen, followed by a steady decrease in RMSD until the completion of the simulation. In contrast to AChE_apo, the AChE in this complex had a notable deviation pattern (0.17 nm) that markedly deviated from the overall stability shown in the AChE_Fumitremorgin C complex (Fig 3a, Table 3). The AChE_Hericenone J complex initially increased RMSD within the first 45 ns. Subsequently, there was an extended period

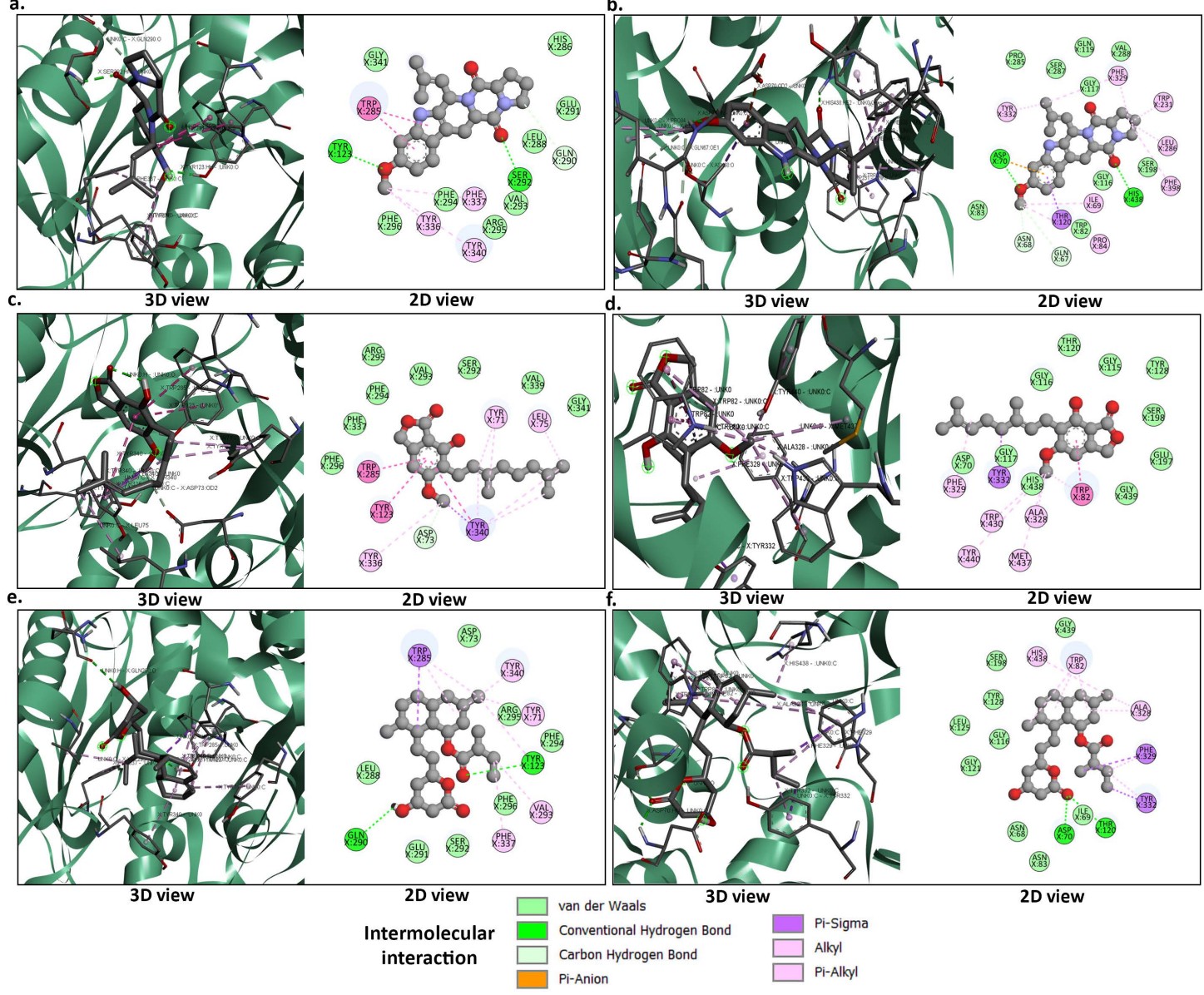

**Fig 2. Post-docking analysis of the fungal metabolites and receptor proteins.** The intermolecular interactions of the AChE_Fumitremorgin C. **(a)**, BuChE_Fumitremorgin C **(b)**, AChE_Hericenone J **(c)**, BChE_Hericenone J **(d)**, AChE_Lovastatin (e), and BuChE_Lovastatin (f) are depicted in ribbon view, where the fungal metabolites are shown in stick view, and the intermolecular interactions are depicted in 2D view.

of stability with minimal fluctuations throughout the simulation. The fluctuation pattern maintained its consistency throughout the entire 100 ns simulation. It is worthwhile that the AChE within this complex exhibited a deviation pattern (<0.1 nm) compared to the AChE_apo (0.17 nm), which was significantly different from the broader stability profile of the AChE_Hericenone J complex. The analysis of these complexes revealed subtle differences in the degree of structural variations observed during stable states. The AChE_Lovastatin complex demonstrated an initial increase in RMSD during the first 13 ns (0.28 nm), followed by an additional spike at 22 ns (0.29 nm). This was followed by an extended stability phase with limited fluctuations during the simulation. The AChE inside this complex demonstrated a deviation pattern (<0.1 nm)

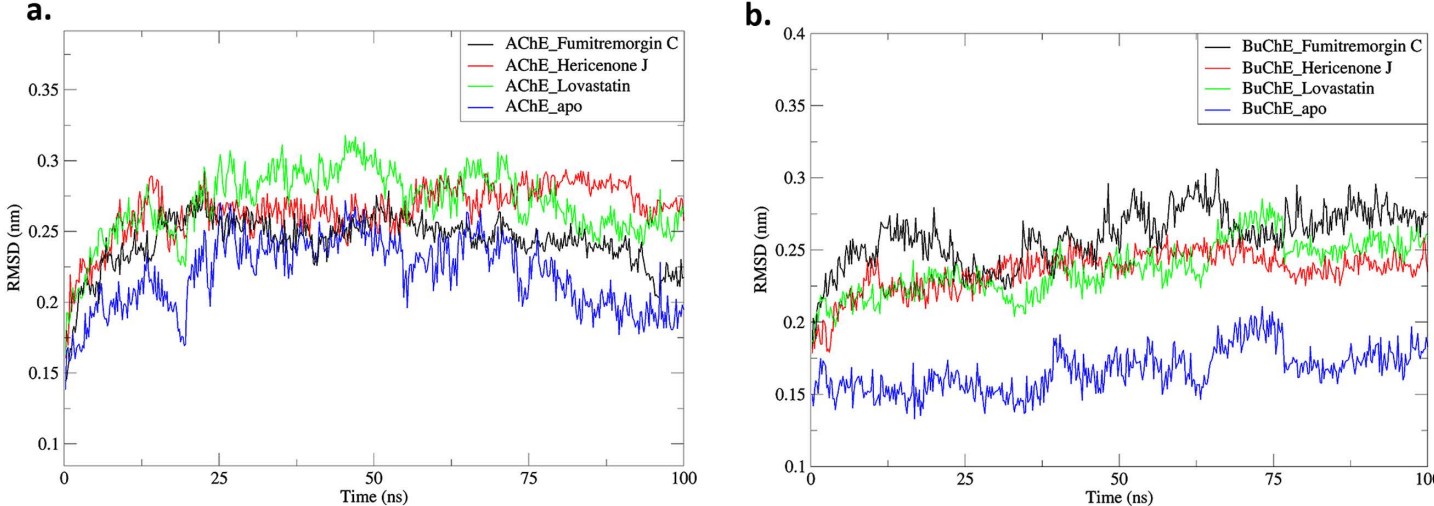

**Fig 3. The RMSD analysis of the AChE-ligand.** (a) and BuChE-ligand (b) complexes.

**Table 3. The RMSD, RMSF, ROG, and SASA analysis of the AChE-ligand and BuChE-ligand complexes.**

| Docked complex | Average RMSD±SD (nm) | Average RMSF (nm) ± SD | Average ROG (nm) ± SD | Average SASA (nm2) ± SD |
|---|---|---|---|---|
| AChE_Fumitremorgin C | 0.24±0.19 | 0.11±0.08 | 2.33±0.01 | 223.89±3.71 |
| AChE_Hericenone J | 0.27±0.19 | 0.12±0.09 | 2.34±0.01 | 219.49±3.07 |
| AChE_Lovastatin | 0.27±0.02 | 0.09±0.09 | 2.34±0.01 | 226.26±3.59 |
| BuChE_Fumitremorgin C | 0.25±0.19 | 0.07±0.07 | 2.34±0.01 | 225.61±2.91 |
| BuChE_Hericenone J | 0.26±0.01 | 0.24±0.06 | 2.33±0.01 | 223.41±3.31 |
| BuChE_Lovastatin | 0.24±0.18 | 0.12±0.07 | 2.32±0.01 | 223.2±3.61 |

in contrast to AChE_apo (0.17 nm), which is markedly different from the larger stability profile of the AChE_Lovastatin complex. The complexes' analyses showed nuanced differences in the extent of structural alterations found during stable states. Although these complexes showed notable stability, the AChE_ Hericenone J complex showed a longer extended period of constant and low-level variations, implying a stronger interaction profile along with a higher increased binding affinity (Fig 3a, Table 3).

The structural stability of BuChE in complexes with various fungal metabolites exhibited distinct differences in RMSD patterns. In the BuChE_Fumitremorgin C complex, the RMSD exhibited a significant increase during the first 13 ns (0.27 nm), followed by a further rise at 48 ns (29 nm) and 65 ns (0.30 nm). In comparison to the BuChE_apo form, the BuChE in this complex had a minimum deviation pattern (0.08 nm), signifying the least change from the overall stability found in the BuChE_Fumitremorgin C complex (Fig 3b, Table 3). Similarly, the BuChE_Hericenone J complex showed an early rise in RMSD at 16 ns (0.24 nm), followed by further fluctuations at 40 ns (0.25 nm). However, sudden fluctuations occurred around 72 ns (0.28 nm), followed by reduced fluctuations and stability for the rest of the simulation time frame. The inconsistent nature of its fluctuation pattern across the 100 ns simulation depicts a deviated structural integrity. Notably, the BuChE in this complex displayed a deviation pattern of less than 0.1 nm compared to the BuChE_apo form (0.17 nm), which makes a considerable difference to the larger stability profile demonstrated by the BuChE_Hericenone J complex. The BuChE_Lovastatin complex had an initial RMSD increase of 0.24 nm within the first 10 ns, followed by an additional rise of 0.21 nm at 41 ns. Subsequently, the complex demonstrated an extended period of stability with a couple

fluctuations. The BuChE in this complex exhibited a deviation pattern of less than 0.1 nm, but the BuChE_apo form had a deviation of 0.17 nm. This deviation pattern significantly differed from the overall stability profile of the BuChE_Lovastatin complex (Fig 3b, Table 3).

**3.5.2. RMSF.** The AChE_Fumitremorgin C, AChE_Hericenone J, AChE_Lovastatin, BuChE_Fumitremorgin C, BuChE_Hericenone J, and BuChE_Lovastatin demonstrated notable patterns in their complexes throughout a 100 ns simulation timeframe. The mean RMSF values for AChE_Fumitremorgin C, AChE_Hericenone J, AChE_Lovastatin, BuChE_Fumitremorgin C, BuChE_Hericenone J, and BuChE_Lovastatin were determined to be 0.11±0.08 nm, 0.12±0.09 nm, 0.09±0.09 nm, 0.07±0.07 nm, 0.24±0.06 nm, and 0.12±0.07 nm, respectively (Fig 4, Table 3).

RMSF analysis was used to evaluate the flexibility of residues in the AChE protein during interactions with various ligands. The analysis demonstrated unique fluctuation patterns in certain residue regions based on the ligand associated with the enzyme. Significant variations were noted in the residue ranges 87–105, 255–260, 380–392, 488–495, and 530–542 for the AChE_Fumitremorgin C complex. In the AChE_Hericenone J complex, variations were observed in the residues ranging from 255 to 268, 285–293, 485–498, and 524–542. Lastly, the AChE_Lovastatin complex exhibited fluctuations in the residue ranges 255–262, 490–500, and 529–542 (Fig 4a, Table 3). Among the three complexes, the AChE_Lovastatin complex showed the least overall fluctuations when compared to the AChE_Fumitremorgin C and AChE_Hericenone J complexes. A detailed examination of residue fluctuations indicated that certain residues displayed significantly lower mobility in specific ligand-bound states. In the AChE_Fumitremorgin C complex, residues GLY-81, ARG-107, ARG-164, GLY-260, GLU-291, ASP-383, PRO-491, PRO-494, and THR-542 showed elevated fluctuations. Similarly, in the AChE_Hericenone J complex, residues ARG-106, ALA-166, THR-261, GLN-290, ASP-383, PRO-491, and LEU-539 exhibited increased flexibility. In contrast, the residues ARG-106, SER-163, PRO-258, PRO-494, and LEU-539 showed more noticeable variations in the AChE_Lovastatin complex (Fig 4a, Table 3).

An RMSF analysis was conducted to evaluate the flexibility of residues in the BuChE during interactions with various ligands. Significant variations in the BuChE_Fumitremorgin C complex were noted in the residue ranges of 40–60, 110–210, 240–275, 340–480, 410–460, and 475–529. In the BuChE_Hericenone J complex, fluctuations were observed in the residues ranging from 255 to 268, 285–293, 485–498, and 524–542. Finally, the BuChE_Lovastatin complex showed variations in the residue ranges of 255–262, 490–500, and 529–542. The analysis of BuChE protein residues in complex

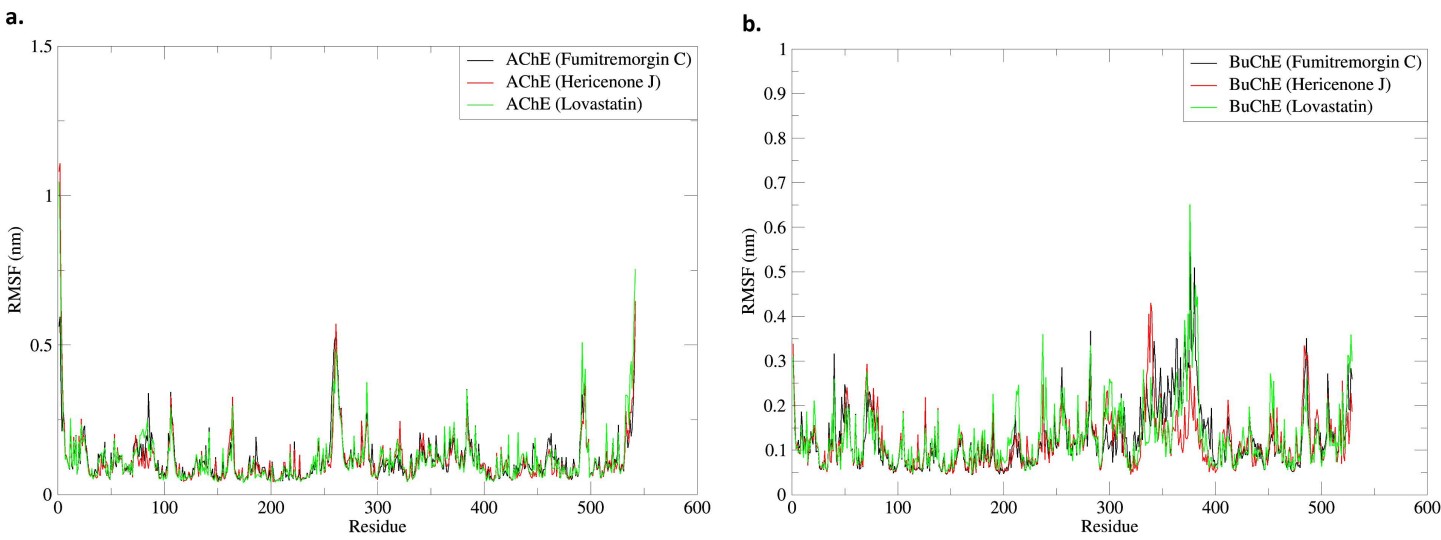

**Fig 4. The RMSF analysis of the AChE-ligand.** (a) and BuChE-ligand (b) complexes.

with various ligands showed differences in residue flexibility (Fig 4b, Table 3). Among the three complexes, the BuChE_ Lovastatin complex showed the least overall fluctuations compared to the AChE_Fumitremorgin C and BuChE_Hericenone J complexes. An in-depth analysis of residue fluctuations revealed that particular residues exhibited notably reduced mobility in certain ligand-bound states. In the BuChE_Fumitremorgin C complex, the residues GLY-39, LEU-49, PRO-74, PHE-76, LYS-105, LEU-125, GLU-137, LYS-190, PRO-211, LEU-236, ARG-254, PRO-281, GLN-311, ALA-328, ALA-334, ASN-341, GLU-363, PHE-371, ASP-375, GLN-380, ASN-485, THR-505, LEU-514, and SER-524 exhibited increased fluctuations. In the BuChE_Hericenone J complex, the residues GLY-39, GLN-71, LYS-190, SER-213, TYR-237, ARG-240, GLU-255, PRO-269, TYR-282, LEU-299, TYR-332, LYS-339, SER-362, HIS-372, ASP-375, TRP-376, GLN-380, GLU-383, ARG-452, and PHE-521 shown more pronounced flexibility. In the BuChE_Lovastatin complex, the residues GLY-39, LYS-51, ASN-106, LEU-125, ARG-135, TYR-237, ARG-240, GLU-255, PRO-269, TYR-282, GLY-296, LEU-299, TYR-332, LYS-339, ASP-375, GLN-380, GLU-383, ASN-485, and PHE-521 exhibited elevated fluctuations (Fig 4b, Table 3).

**3.5.3. Radius of gyration.** Over the course of a 100 ns simulation period, the complexes of AChE_Fumitremorgin C, AChE_Hericenone J, AChE_Lovastatin, BuChE_Fumitremorgin C, BuChE_Hericenone J, and BuChE_Lovastatin displayed unique structural patterns. The mean ROG values for these compounds were documented as 2.33±0.01 nm, 2.34±0.01 nm, 2.34±0.01 nm, 2.34±0.01 nm, 2.33±0.01 nm, and 2.32±0.01 nm, respectively (Fig 5, Table 3). In comparison to AChE_apo, the AChE structure in the AChE_Fumitremorgin C complex exhibited no alteration (2.33 nm), but slight structural variations (0.01 nm) were seen in the AChE_Hericenone J and AChE_Lovastatin complexes (Fig 5a, Table 3). Similarly, small changes (0.01 nm) were observed in the BuChE_Fumitremorgin C and BuChE_ Lovastatin complexes. However, BuChE_Fumitremorgin C did not show any structural changes (Fig 5b, Table 3).

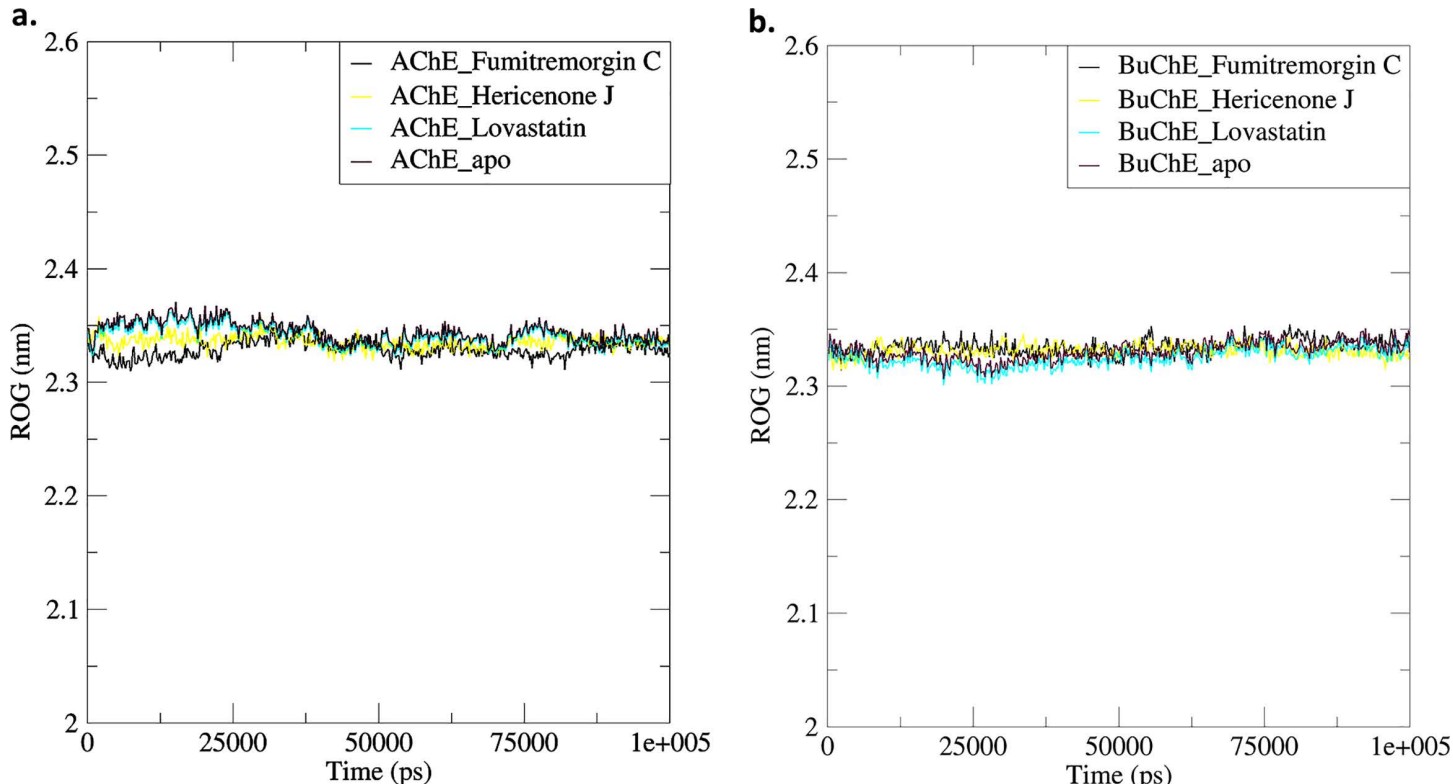

**Fig 5. The radius of gyration analysis of the AChE-ligand.** (a) and BuChE-ligand (b) complexes.

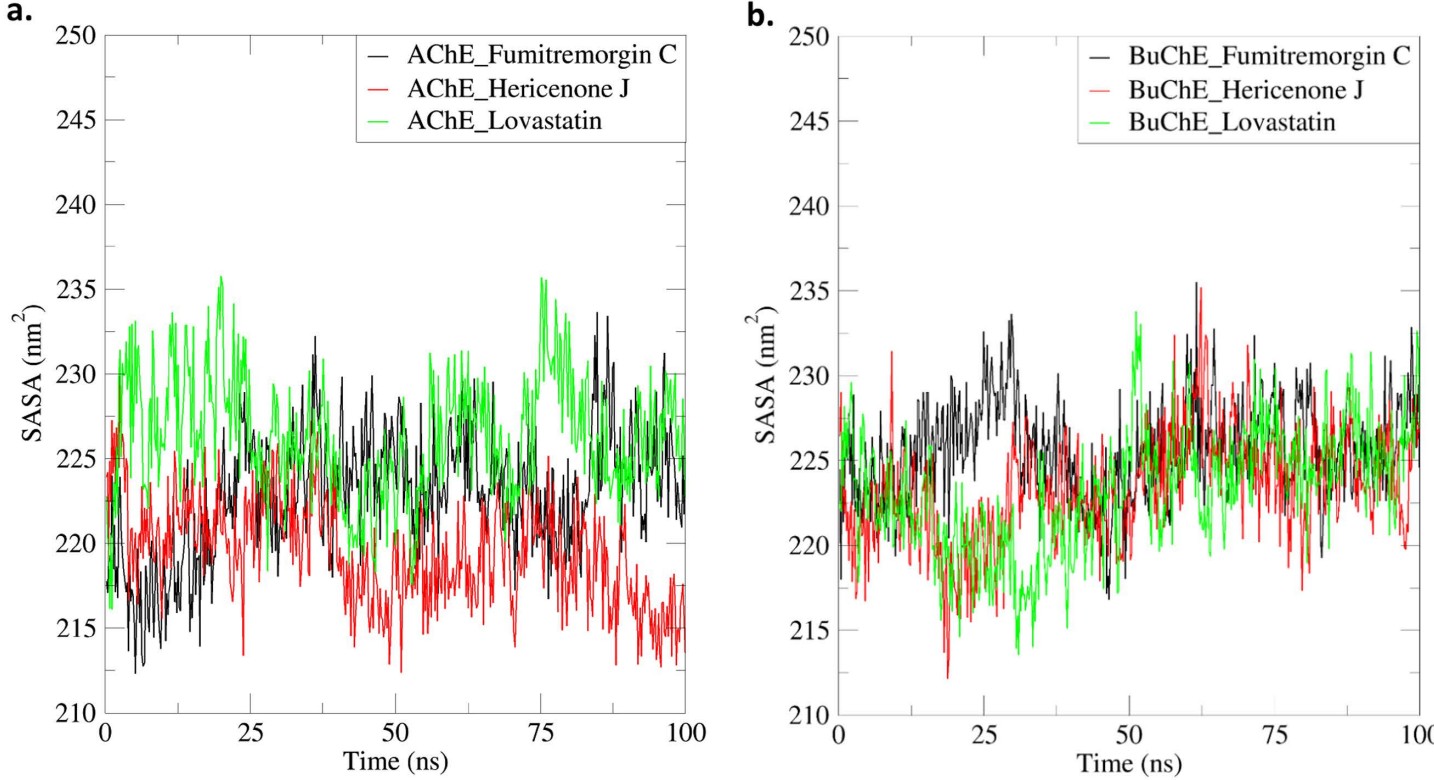

**Fig 6. The SASA analysis of the AChE-ligand.** (a) and BuChE-ligand (b) complexes.

**3.5.4. SASA.** The SASA is crucial for understanding protein-ligand complexes' structural stability and dynamic behaviour. In the 100 ns simulation trajectories, the AChE_Fumitremorgin C, AChE_Hericenone J, AChE_ Lovastatin, BuChE_Fumitremorgin C, BuChE_Hericenone J, and BuChE_Lovastatin showed distinct SASA patterns. The mean SASA values for the AChE-bound complexes—AChE_Fumitremorgin C, AChE_Hericenone J, and AChE_Lovastatin—were reported as 222.89 ± 3.71 nm², 219.49 ± 3.07 nm², and 226.26 ± 3.59 nm², respectively. The BuChE complexes—BuChE_ Fumitremorgin C, BuChE_Hericenone J, and BuChE_Lovastatin—exhibited mean SASA values of 225.61 ± 2.91 nm², 223.41 ± 3.31 nm², and 223.2 ± 3.61 nm², respectively (Fig 6, Table 3).

All ligand-bound AChE and BuChE complexes had a reduced solvent-exposed surface area relative to their apo forms, with SASA values below 236.41 nm² for AChE and 237.81 nm² for BuChE. Subsequent analysis of the SASA variations across time revealed specific stages of structural variations in certain complexes. The AChE_Fumitremorgin C complex significantly fluctuated SASA values, especially between 1–36 ns and 81–93 ns. The AChE_Lovastatin complex exhibited notable changes at many time intervals, notably between 1–11 ns, 24–30 ns, 39–55 ns, 68–71 ns, and 82–92 ns (Fig 6a, Table 3). The BuChE complexes exhibited significant variations in SASA during the simulation. The BuChE_Hericenone J complex experienced significant fluctuations at 1–10 ns, 32–38 ns, and 68–81 ns, whereas BuChE_Fumitremorgin C showed notable variations between 25–59 ns, 61–71 ns, and 74–90 ns. The BuChE_Lovastatin complex showed significant SASA changes at various important time points, specifically at 1–6 ns, 13–20 ns, 29–36 ns, 49–67 ns, and 75–99 ns (Fig 6b, Table 3).

**3.4.5. Hydrogen bonds.** Hydrogen bonds are crucial in molecular recognition by facilitating specific and directed interactions between proteins and their ligands. As a post-simulation analysis, the number of hydrogen bonds was evaluated for the AChE_Fumitremorgin C, AChE_Hericenone J, AChE_Lovastatin, BuChE_Fumitremorgin C,

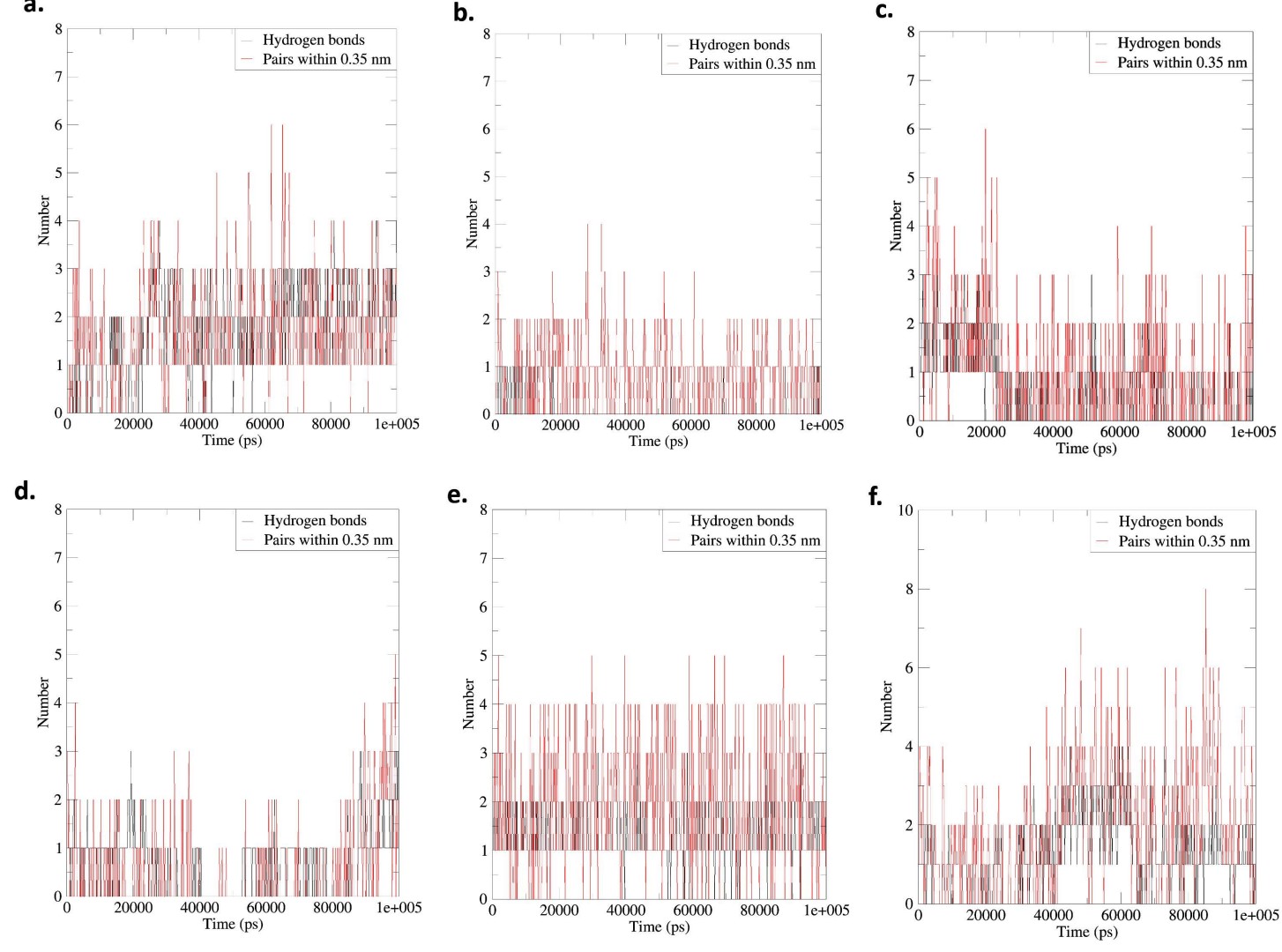

**Fig 7. The hydrogen bond analysis of the AChE_Fumitremorgin C. (a)**, AChE_Hericenone J **(b)**, AChE_Lovastatin **(c)**, BuChE_Fumitremorgin C **(d)**, BuChE_Hericenone J **(e)**, and BuChE_Lovastatin **(f)**.

BuChE_Hericenone J, and BuChE_Lovastatin complex at different simulation trajectories. The AChE_Fumitremorgin C (GLU-291, and SER-292), and AChE_Lovastatin (TYR-123, and SER-292) complexes each had two hydrogen bonds, but AChE_Hericenone J (GLY-125, ALA-126, and TYR-132), had three (Figs 7 and 8). Conversely, the BuChE_Lovastatin and BuChE_Hericenone J complexes established three (ILE-69, ASP-70, and SER-287) and four (TRP-82, THR-120, THR-122, and TYR-128) hydrogen bonds, respectively. The BuChE_Fumitremorgin C complex exhibited minimal hydrogen bonding interaction, with just one (THR-120) hydrogen bond detected (Figs 7 and 9).

In the 25 ns simulation trajectory, five hydrogen bonds were identified for the AChE_Fumitremorgin C complex (GLN-290, SER-292, VAL-293, PHE-294, and ARG-295). In contrast, the AChE_Hericenone J complex exhibited four hydrogen bonds (TRP-85, GLY-119, ALA-126, and TYR-132), and the AChE_Lovastatin complex displayed only one hydrogen bond (SER-292) (Figs 7 and 8). Conversely, the BuChE_Hericenone J complexes established four (TRP-82, ASN-83, ASP-70, and GLN-71) hydrogen bonds, respectively. The BuChE_Fumitremorgin C complex exhibited minimal hydrogen bonding

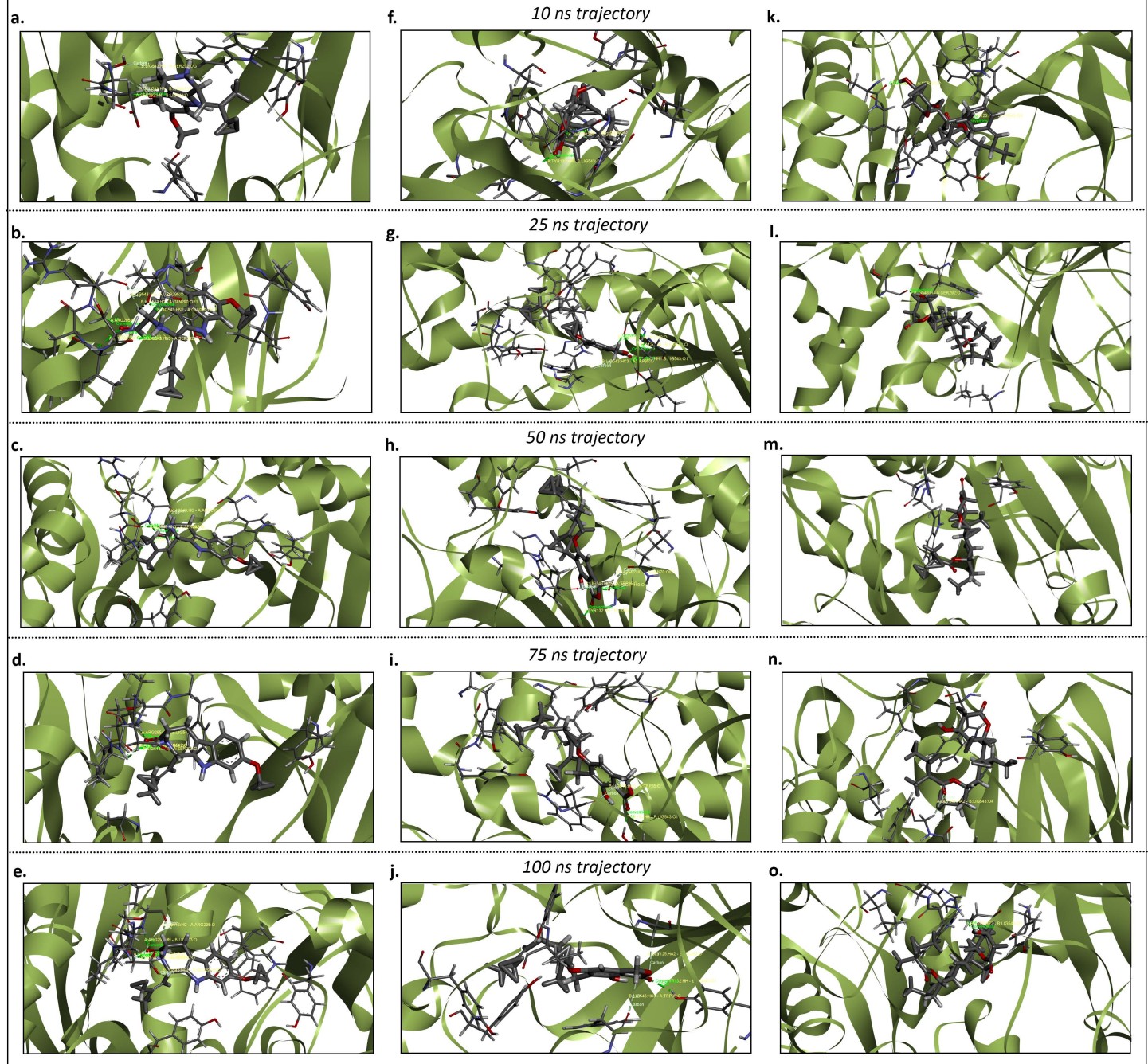

**Fig 8. The hydrogen bond analysis of the AChE_Fumitremorgin C. (a-e)**, AChE_Hericenone J **(g-j)**, AChE_Lovastatin (i-o) complexes at different simulation trajectories including 1 ns, 25 ns, 50 ns, 75 ns, and 100 ns.

interaction, with just one (GLY-439) hydrogen bond detected. However, the AChE_Lovastatin complex had no hydrogen bonds (Figs 7 and 9).

In the 50 ns simulation trajectory, two hydrogen bonds were identified in the AChE_Fumitremorgin C complex (PHE-294 and ARG-295), and four in the AChE_Hericenone J complex (GLN-70, TRP-85, GLY-119, and TYR-132); however,

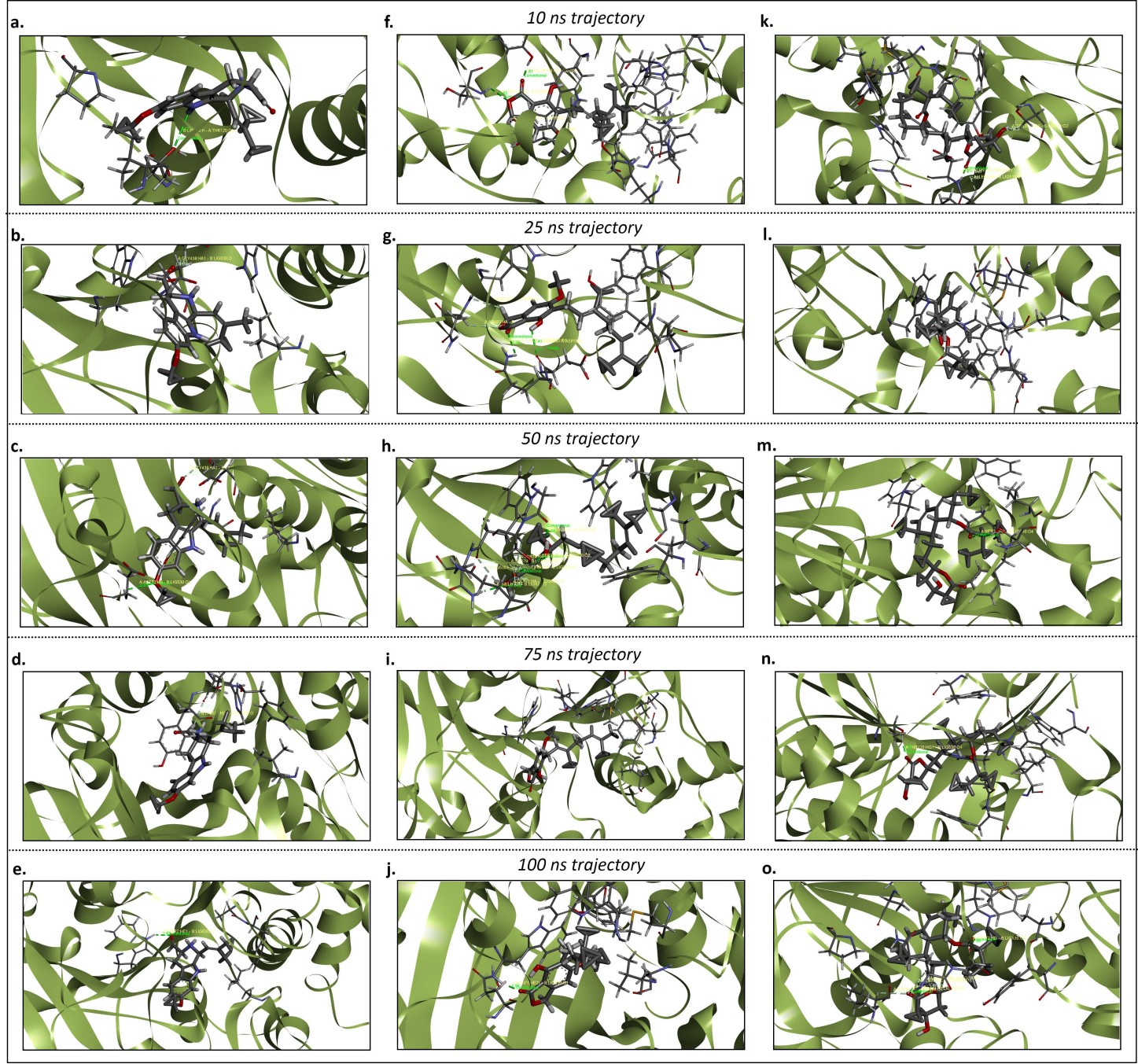

**Fig 9. The hydrogen bond analysis of the BuChE_Fumitremorgin C. (a-e)**, BuChE_Hericenone J **(g-j)**, BuChE_Lovastatin (i-o) complexes at different simulation trajectories including 1 ns, 25 ns, 50 ns, 75 ns, and 100 ns.

the AChE_Lovastatin complex exhibited no hydrogen bonds (Figs 7 and 8). The BuChE_Fumitremorgin C complex (ASP-70, and GLY-439) had two hydrogen bonds, the BuChE_Hericenone J complex (GLN-71, GLY-78, SER-79, TRP-82, and ASN-83) had five, and the BuChE_Lovastatin complex had only one (SER-72) (Figs 7 and 9).

In the 75 ns simulation trajectory, the AChE_Fumitremorgin C complex demonstrated four hydrogen bonds (SER-292, VAL-293, PHE-294, and ARG-295), while the AChE_Hericenone J complex established two (TRP-85 and TYR-132). The AChE_Lovastatin complex exhibited a solitary hydrogen bond (GLY-341) (Figs 7 and 8). Additionally, a hydrogen bond was seen in both the BuChE_Fumitremorgin C (HIS-438) and BuChE_Lovastatin (THR-120) complexes; however, the BuChE_Hericenone J complex had no hydrogen bonds (Figs 7 and 9).

During the 100 ns simulation trajectory, the AChE_Fumitremorgin C complex demonstrated the formation of four hydrogen bonds involving SER-292, VAL-293, PHE-294, and ARG-295. In contrast, the AChE_Hericenone J complex established three hydrogen bonds with TRP-85, GLY-125, and TYR-132 (Figs 7 and 8). The AChE_Lovastatin complex had one hydrogen bond (HIS-286). A hydrogen bond was seen in both the BuChE_Fumitremorgin C (TRP-82) and BuChE_Hericenone J (GLN-71) complexes, while the BuChE_Lovastatin complex exhibited three hydrogen bonds (ILE-69, GLN-71, and SER-72) (Figs 7 and 9).

### 3.5. MMPBSA

This research also explores the binding energy elements of AChE and BuChE as they interact with Fumitremorgin C, Hericenone J, and Lovastatin. The analysis reveals significant disparities in the energetic profiles of these complexes, as evidenced by their total binding energies of −1420.63 kcal/mol, −1677.95 kcal/mol, −1378.02 kcal/mol, −1070.92 kcal/mol, −1484.06 kcal/mol, and −1618 kcal/mol for AChE_Fumitremorgin C, AChE_Hericenone J, AChE_Lovastatin, BuChE_Fumitremorgin C, BuChE_Hericenone J, and BuChE_Lovastatin (Table 4). The interaction between AChE and Fumitremorgin C produced the following energy components: ΔVDWAALS: −1553.41 kcal/mol, ΔEEL: −11221.33 kcal/mol, ΔEGB: 11566.88 kcal/mol, ΔESURF: −212.76 kcal/mol, ΔGGAS: −12774.72 kcal/mol, ΔGSOLV: 11354.09 kcal/mol. Likewise, ΔVDWAALS (−2239.86 kcal/mol), ΔEEL (−630.95 kcal/mol), ΔEGB (1452.74 kcal/mol), ΔESURF (−259.83 kcal/mol), ΔGGAS (−2870.8 kcal/mol), and ΔGSOLV (1192.89 kcal/mol) were the values of AChE binding with Hericenone J. The energy components for the AChE-Lovastatin complex were as follows: ΔVDWAALS (−1986.43 kcal/mol), ΔEEL (−337.16 kcal/mol), ΔEGB (1190.87 kcal/mol), ΔESURF (−245.26 kcal/mol), ΔGGAS (−2323.54 kcal/mol), and ΔGSOLV (945.48 kcal/mol) (Table 4).

The interaction of BuChE with Fumitremorgin C exhibited the following binding energy components: ΔVDWAALS (−1683.03 kcal/mol), ΔEEL (−5601.22 kcal/mol), ΔEGB (6435.74 kcal/mol), ΔESURF (−222.34 kcal/mol), ΔGGAS (−7284.35 kcal/mol), and ΔGSOLV (6213.4 kcal/mol). For the BuChE-Hericenone J complex, the energy values were as follows: ΔVDWAALS (−1878.16 kcal/mol), ΔEEL (−716.48 kcal/mol), ΔEGB (1343.74 kcal/mol), ΔESURF (−233.18 kcal/mol), ΔGGAS (−2594.63 kcal/mol), and ΔGSOLV (1110.59 kcal/mol). Finally, the BuChE-Lovastatin complex demonstrated binding energy values of ΔVDWAALS (−2334.09 kcal/mol), ΔEEL (−279.47 kcal/mol), ΔEGB (1270.92 kcal/mol), ΔESURF (−275.32 kcal/mol), ΔGGAS (−2613.56 kcal/mol), and ΔGSOLV (995.55 kcal/mol) (Table 4).

**Table 4. The MMPBSA including ΔVDWAALS, ΔEEL, ΔEGB, ΔESURF, ΔGGAS, and ΔGSOLV analysis of the the AChE-ligand and BuChE-ligand complexes.**

| Docked complex | ΔVDWAALS | ΔEEL | ΔEGB | ΔESURF | ΔGGAS | ΔGSOLV | Total |
|---|---|---|---|---|---|---|---|
| AChE_Fumitremorgin C | −1553.41 | −11221.3 | 11566.88 | −212.76 | −12774.7 | 11354.09 | −1420.63 |
| AChE_Hericenone J | −2239.86 | −630.95 | 1452.74 | −259.83 | −2870.8 | 1192.89 | −1677.95 |
| AChE_Lovastatin | −1986.43 | −337.16 | 1190.87 | −245.26 | −2323.54 | 945.48 | −1378.02 |
| BuChE_Fumitremorgin C | −1683.03 | −5601.22 | 6435.74 | −222.34 | −7284.35 | 6213.4 | −1070.92 |
| BuChE_Hericenone J | −1878.16 | −716.48 | 1343.74 | −233.18 | −2594.63 | 1110.59 | −1484.06 |
| BuChE_Lovastatin | −2334.09 | −279.47 | 1270.92 | −275.32 | −2613.56 | 995.55 | −1618 |

## 3.6 Principal component and Gibbs free energy analysis

The most significant distinctions among the complexes were examined by arranging the main components into eigenvectors prioritized based on their variability. Scatter plots have been developed for the AChE_Fumitremorgin C, AChE_Hericenone J, AChE_Lovastatin, BuChE_Fumitremorgin C, BuChE_Hericenone J, and BuChE_Lovastatin complexes, displaying the simulated trajectories in a two-dimensional space defined by the first two eigenvectors (PC1 and PC2). PCA exhibited self-directed movements within the AChE and BuChE and their ligand complexes. The PCA-based Gibbs free energy analysis indicated that the complex of AChE with Fumitremorgin C exhibited the lowest energy conformations within a distinct energy basin, spanning the reaction coordinates from −4.96 to 3.72 on PC1 and −2.85 to 3.82 on PC2. In comparison, the complex of AChE with Hericenone J showed two lowest energy conformations where the more significant number of conformations occupy the energy basin between − 4.09 to 5.73 on PC1 and −2.93 to 3.18 on PC2. The AChE with Lovastatin exhibited a distinct lowest energy basin ranging from −4.15 to 6.45 on PC1 and −5.55 to 4.77 on PC2 (**Fig 10**). The PCA-based Gibbs free energy analysis revealed that the BuChE and Fumitremorgin C complex exhibited the lowest energy conformations inside a specific energy basin, covering the reaction coordinates of −5.71 to 2.89 on PC1 and −3.72 to 5.73 on PC2. Conversely, the BuChE with Hericenone J had the two lowest energy conformations. A larger quantity of conformations resides inside the energy basin ranging from −3.23 to 4.76 on PC1 and −3.62 to 3.25 on PC2. The BuChE with Lovastatin exhibited a distinctive lowest energy basin ranging from −6.20 to 3.39 on PC1 and −4.81 to 3.90 on PC2 (**Fig 10**).

The Gibbs free energy landscape was calculated to assess the stability of the AChE-ligand and BuChE-ligand complexes, utilizing two principal components (PC1 and PC2). Subsequent analysis of the AChE-ligand complexes revealed that the AChE_Lovastatin complex exhibited more low-energy basins than the AChE_Fumitremorgin C and AChE_Hericenone J complexes. The energy ranges for the AChE_Fumitremorgin C, AChE_Hericenone J, and AChE_Lovastatin complexes were calculated as 0–5.85 kJ/mol, 0.29–6.98 kJ/mol, and 0–6.98 kJ/mol, respectively (**Fig 11**). Conversely, the study of the BuChE-ligand complexes indicated that the BuChE_Hericenone J complex exhibited an increased low-energy basins compared to the BuChE_Fumitremorgin C and BuChE_Lovastatin complexes. The energy ranges for the BuChE_ Fumitremorgin C, BuChE_Hericenone J, and BuChE_Lovastatin complexes were calculated as 0–5.85 kJ/mol, 0.29–6.98 kJ/mol, and 0–6.98 kJ/mol, respectively (**Fig 11**).

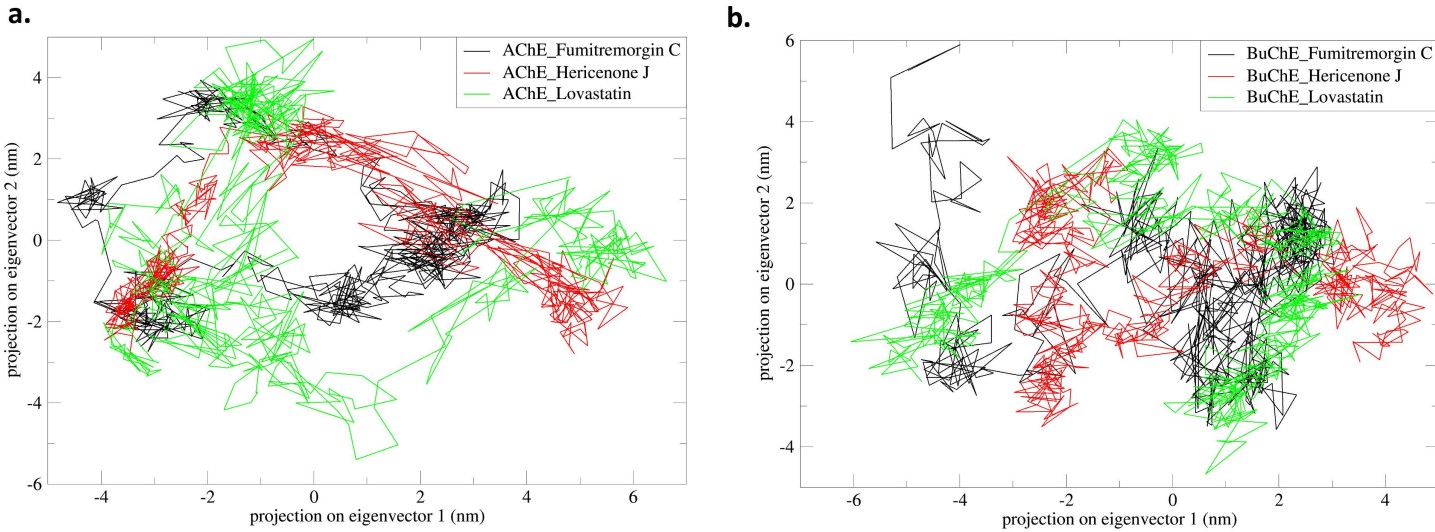

**Fig 10. The principal component analysis of the AChE-ligand.** (a) and BuChE-ligand (b) complexes.

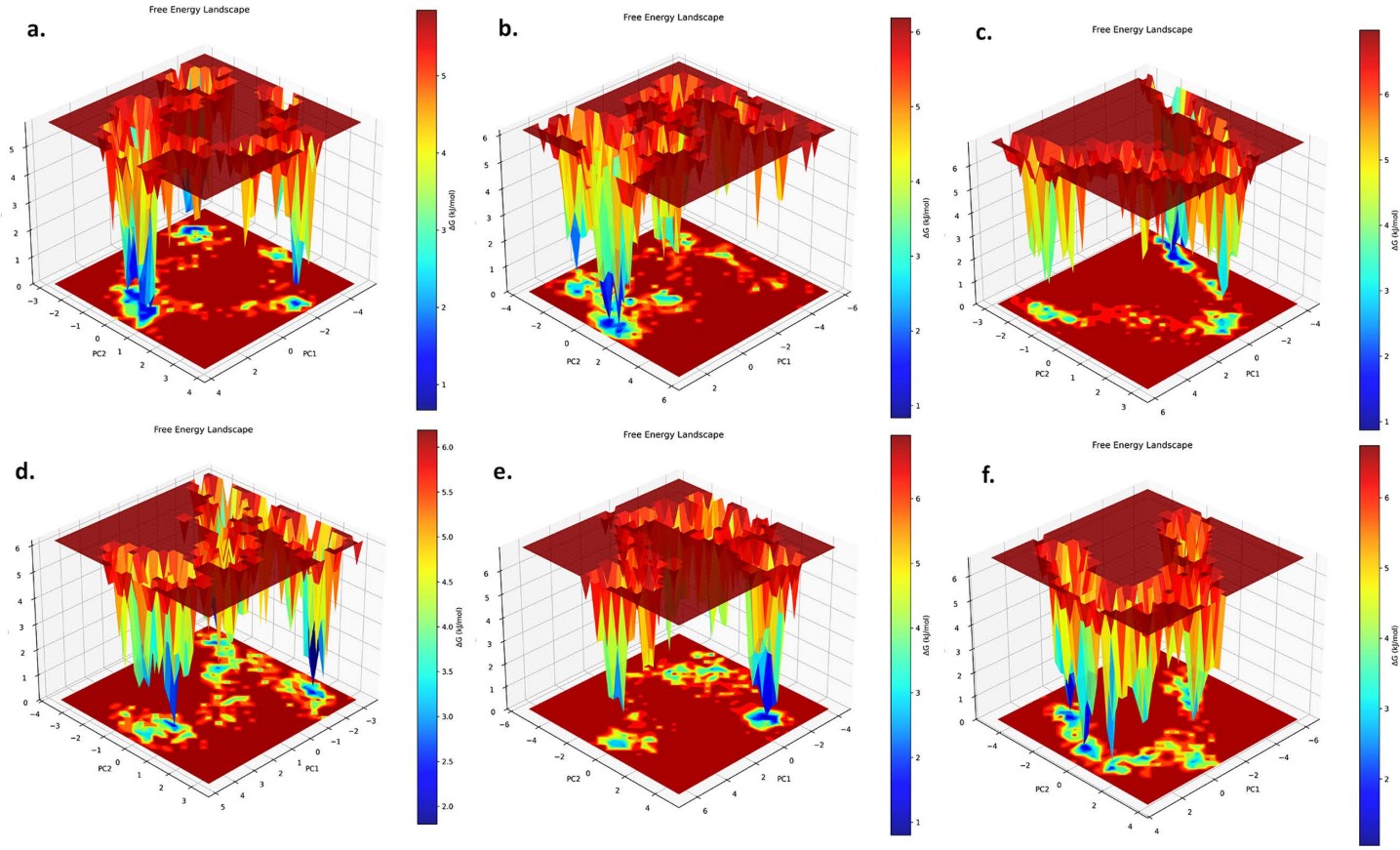

**Fig 11. The Gibbs free energy analysis of the AChE_Fumitremorgin C. (a)**, AChE_Hericenone J **(b)**, AChE_Lovastatin **(c)**, BuChE_Fumitremorgin C **(d)**, BuChE_Hericenone J **(e)**, and BuChE_Lovastatin (f) complexes.

## 4. Discussion

Even with the existing conventional alternatives for AD, it is still important to explore novel therapies. Cholinesterase inhibitors and NMDA receptor antagonists are two examples of current medications that provide merely short-term cognitive improvements and fail to reverse the progression of the disease [54–56]. The restricted effectiveness of conventional treatments, along with the significant failure rates in the development of novel drugs due to inadequate efficacy and negative side effects, has heightened need for the advancement of the next wave of AD therapies [55]. Additionally, the intricate and varied factors contributing to the onset of AD, present considerable hurdles for the development of novel drugs [57,58]. Due to the complex nature of AD, marked by amyloid-beta plaques, tau protein tangles, and neuroinflammation, there is an urgent need for novel therapies that directly address these fundamental processes. Also, the restricted effectiveness of single-target medications, research has transitioned to the development of dual- or multi-target inhibitors [59–61]. These inhibitors augment therapy by targeting several sites with synergistic effects, enhancing effectiveness, diminishing negative effects, and decreasing therapeutic dosages while minimizing drug-drug interactions [62]. Cholinesterase inhibitors are essential in AD treatment since cholinergic dysfunction is associated with Aβ and tau pathology [63]. It has been shown that inhibiting both AChE and BuChE may improve cognitive function and reduce the production of Aβ plaque [19]. Recent developments in genomics, biomarker research, and precision medicine present new possibilities for more effective, disease-modifying treatments, highlighting the importance of prioritizing drug discovery [55].

In this study, we employed CADD to develop AD drugs that target both the AChE and BuChE at a time, utilizing fungal secondary metabolites obtained from the MeFSAT. Subsequent PK analysis revealed that all the metabolites demonstrated high gastrointestinal (GI) absorption, indicating favorable oral bioavailability [64,65], and was also blood-brain barrier (BBB) permeant, suggesting potential for neurological or neuroprotective applications [34,35,66]. Furthermore, none of the compounds were identified as P-glycoprotein (P-gp) substrates, reducing the likelihood of active efflux and enhancing intracellular drug retention, which could improve therapeutic efficacy [65]. All bioactive compounds complied with Lipinski's Rule of Five without any violations, indicating favorable oral bioavailability. Additionally, all metabolites met the criteria for Ghose, Veber, Egan, and Muegge filters, further supporting their drug-likeness and pharmacokinetic feasibility [67]. The bioavailability score was 0.55 for all compounds, suggesting moderate oral absorption potential [68,69].

The results indicated that all fungal metabolites were non-hepatotoxic, suggesting a low risk of liver toxicity, and is crucial for drug metabolism and systemic tolerance [70,71]. All metabolites, with the exception of Lovastatin, which was identified as carcinogenic, were categorised as non-carcinogenic, suggesting a small likelihood of cancer-related side effects [70]. Moreover, none of the metabolites demonstrated mutagenicity or cytotoxicity, hence reinforcing their potential safety for therapeutic use. The results suggested that, aside from the concerns regarding Lovastatin's potential carcinogenic effects, the other fungal metabolites showed promising toxicity profiles, considering them as strong candidates for drug optimization and subsequent preclinical assessment [70,71].

The Chiv5 index, which spans from 1.40 to 3.82, reflects different levels of molecular complexity. Higher values imply greater branching and size, which may improve interactions with targets [72]. The bcutm1 values, ranging from 3.90 to 4.12, indicate minor variations in atomic mass distribution, potentially affecting molecule stability and binding efficacy [72]. Descriptors such as MRVSA9 (5.91–22.72) and MRVSA6 (23.80–46.06) underscore the differences in molecular polarisability and steric effects, which are essential for enhancing ligand-receptor interactions [72]. The PEOEVSA5 range of 11.65–45.92 signifies varying electrostatic potentials, influencing molecule recognition and binding affinity. The GATSv4 values (0.98–1.24) and J index (1.46–2.15) indicate variations in molecular geometry and topological structure, potentially affecting target accessibility [72]. Additionally, Diametert varies from 7.00 to 13.00, reflecting differences in molecular size, which may affect the ability to fit into binding pockets [72]. Finally, pIC50 values range from 4.41 to 5.21, with Lovastatin (5.21) demonstrating the highest potency, making it a promising lead compound. These findings provide a structural basis for optimizing fungal metabolites as potential drug candidates through targeted modifications [72]. Following the evaluation of docking scores, pharmacokinetic profiles, and QSAR analysis, three fungal metabolites—Fumitremorgin C, Hericenone J, and Lovastatin—were selected for further molecular dynamic simulation.

The post-docking analysis of the fungal metabolites demonstrated strong binding affinities toward both AChE and BuChE, which sometimes prevail over control drugs. Within the compounds evaluated, Fumitremorgin C showed the strongest binding affinity for AChE at −10.0 kcal/mol, along with a notable affinity for BuChE at −9.1 kcal/mol, surpassing galantamine and rivastigmine and approaching the efficacy of donepezil. The enhanced affinity for the receptors suggests its possible involvement in influencing cholinergic neurotransmission, a key factor in treating AD. The Hericenone J, conversely, showed a marginally reduced docking score with AChE (−8.0 kcal/mol) while demonstrating a greater affinity for BuChE (−9.2 kcal/mol). The compound demonstrated comparable binding energies to galantamine and donepezil, exhibiting superior binding affinities to rivastigmine. This indicates that although it might not be as efficient in blocking AChE compared to Fumitremorgin C, it has the potential to act as a stronger BuChE inhibitor. As the progression of AD leads to an increase in BuChE activity, Hericenone J could prove advantageous during the later stages of the condition. Lovastatin demonstrated moderate binding affinities, achieving docking scores of −9.2 kcal/mol for AChE and −8.6 kcal/mol for BuChE. While it demonstrated slightly less potency than Fumitremorgin C in inhibiting AChE, Lovastatin still showed significant promise in addressing both cholinesterases. The compound demonstrated superior binding affinities compared with rivastigmine. Nonetheless, the compound exhibited reduced efficacy compared with galantamine and donepezil. Recognized for its ability to reduce cholesterol, its dual effect on AChE and BuChE indicates a potential neuroprotective function beyond lipid metabolism.

The stability of AChE and BuChE complexes with fungal metabolites—Fumitremorgin C, Hericenone J, and Lovastatin—was evaluated throughout a 100 ns simulation by RMSD analysis. The RMSD values elucidate the structural aberrations of these complexes in comparison to their apo forms. The complexes demonstrated unique RMSD patterns for AChE. The AChE_Fumitremorgin C exhibited an initial increase to 23 ns (0.29 nm), followed by stabilization, while the AChE_Hericenone J showed an early rise before achieving a stable profile. The AChE_Lovastatin demonstrated a fast increase in RMSD during the first 13 ns (0.28 nm), followed by a prolonged stability period. All the AChE-ligand complexes had reduced deviation patterns (<0.1 nm) relative to AChE_apo (0.17 nm), indicating enhanced stability. The AChE_Hericenone J had the greatest stability duration, indicating a superior binding affinity [73–75]. The BuChE_ligand complexes had distinctive RMSD characteristics. The BuChE_Fumitremorgin C exhibited initial fluctuations, reaching a peak at 65 ns (0.30 nm), while the BuChE_Hericenone J showed fluctuating instability at around 72 ns. The BuChE_Lovastatin demonstrated a significant increase at 10 ns (0.24 nm), thereafter maintaining sustained stability. In contrast to the BuChE_apo (0.17 nm), all the BuChE-ligand complexes exhibited negligible variations (<0.1 nm), indicating their general stability [73–75].

The RMSF analysis evaluated the flexibility of AChE and BuChE residues within complexes formed with fungal metabolites during a 100 ns simulation. The average RMSF values for the AChE-ligand complexes varied between 0.09 and 0.12 nm, whereas the BuChE-ligand complexes exhibited more significant fluctuations, especially the BuChE_Hericenone J (0.24 nm). In the case of AChE-ligands, the fluctuations of residues differed among the various complexes. The AChE_Fumitremorgin C demonstrated significant flexibility in the regions of residues 87–105, 255–260, and 380–392, whereas the AChE_Hericenone J displayed variations in the segments 255–268 and 485–498. The AChE_Lovastatin exhibited minimal variation, particularly in the ranges of 255–262 and 490–500 [73–75]. The BuChE-ligand complexes exhibited unique fluctuating patterns. The BuChE_Fumitremorgin C showed oscillations over wider residue ranges, while the BuChE_Hericenone J and BuChE_Lovastatin displayed more confined variations. The BuChE_Lovastatin had the lowest volatility, indicating a more stable interaction [73–75].

The ROG analysis offers valuable insights into the compactness and structural integrity of the AChE-ligand and BuChE-ligand complexes throughout a 100 ns simulation. The observed ROG values for the AChE-ligand complexes were between 2.33 nm and 2.34 nm, whereas the BuChE-ligand complexes showed values ranging from 2.32 nm to 2.34 nm. In comparison to the AChE_apo, the AChE_Fumitremorgin C complex exhibited structural stability without any deviation (2.33 nm). Minor deviations (0.01 nm) were noted in the AChE_Hericenone J and AChE_Lovastatin, indicating subtle structural changes upon ligand binding [73–75]. Likewise, the BuChE-ligand complexes exhibited little changes, suggesting that ligand interaction did not substantially affect structural compactness. The uniformity of ROG values across all complexes indicates stable ligand-protein interactions without significant conformational destabilization. The results underscore the structural integrity of the enzyme-ligand complexes, with negligible variations enhancing the stability of fungal metabolite-bound AChE and BuChE structures throughout the simulation. [73–75].

The SASA analysis offers insights into the structural stability and solvent accessibility of the AChE-ligand and BuChE-ligand complexes over a 100 ns simulation. The average SASA values for the AChE-ligand complexes varied from 219.49 nm² to 226.26 nm², while the BuChE-ligand complexes had SASA values ranging from 223.2 nm² to 225.61 nm². All ligand-bound complexes exhibited lower SASA values compared to their corresponding apo forms, indicating less solvent exposure and increased stability [73–75]. Significant variations were noted during particular periods. The AChE_Fumitremorgin C demonstrated notable SASA changes during the periods of 1–36 ns and 81–93 ns, whereas the AChE_Lovastatin displayed variations at several intervals, specifically 1–11 ns and 82–92 ns. The BuChE-ligand complexes exhibited notable variations, with the BuChE_Hericenone J oscillating between 1–10 ns and 68–81 ns. The BuChE_Lovastatin showed significant variations ranging from 1–6 ns to 75–99 ns. The differences underscore the dynamic nature of the protein-ligand complexes, with structures bound to Lovastatin showing enhanced flexibility [73–75].

Hydrogen bonding is essential for the stability of protein-ligand complexes, as it promotes specific molecular interactions. Analysis following the simulation showed clear differences in hydrogen bond patterns for the AChE-ligand and

BuChE-ligand complexes at various time intervals. The AChE_Hericenone J consistently established the maximum number of hydrogen bonds (3–4) with the AChE-bound ligands across all time frames, engaging residues including GLY-125, TYR-132, and TRP-85. The AChE_Fumitremorgin C demonstrated dynamic hydrogen bonding, establishing up to five bonds at 25 ns, but stabilizing at four by 100 ns. In contrast, the AChE_Lovastatin exhibited little hydrogen bonding, with just one bond occurring at certain intervals (HIS-286 at 100 ns). In the analysis of BuChE-ligand complexes, the BuChE_Hericenone J consistently exhibited robust hydrogen bonding interactions, with 4–5 bonds, while the BuChE_Fumitremorgin C showed the weakest interactions, forming only a single bond at most time points. The BuChE_Lovastatin showed a fair level of stability, consistently holding onto 1–3 hydrogen bonds. The differences observed suggest varying levels of ligand stabilization, with Hericenone J enhancing hydrogen bonding interactions in both AChE-ligand and BuChE-ligand complexes [73–75].

Analysis of binding energy indicated substantial differences in the interactions of AChE and BuChE with Fumitremorgin C, Hericenone J, and Lovastatin. The AChE_Hericenone J had the most robust binding affinity (−1677.95 kcal/mol), followed by the BuChE_Lovastatin (−1618 kcal/mol). The ΔVDWAALS and ΔEEL contributions were significant, with the AChE_Hericenone J exhibiting the greatest ΔVDWAALS value of −2239.86 kcal/mol. The ΔEEL and ΔGSOLV energies significantly influenced interactions. The BuChE_Fumitremorgin C had the lowest binding energy (−1070.92 kcal/mol), indicating a diminished affinity. In summary, Hericenone J and Lovastatin complexes exhibited superior binding compared to Fumitremorgin C, indicating their potential as more stable inhibitors for AChE and BuChE [73–75]. Analysis using principal components highlighted clear differences in the conformations of AChE-ligand and BuChE-ligand complexes. The AChE_Fumitremorgin C complex demonstrated the most stable and lowest energy basin, ranging from −4.96 to 3.72 on PC1 and from −2.85 to 3.82 on PC2. The AChE_Hericenone J and AChE_Lovastatin exhibited the two lowest energy basins, with Lovastatin covering the widest range. In a similar vein, the BuChE_Fumitremorgin C displayed a clearly defined lowest energy basin (−5.71 to 2.89 on PC1, −3.72 to 5.73 on PC2). In contrast, the BuChE_Hericenone J and BuChE_Lovastatin revealed unique conformational clusters, indicating variations in binding stabilities and dynamic behaviors [73–75].

An analysis of the Gibbs free energy landscape for the AChE-ligand and BuChE-ligand complexes revealed that lower free energy values are associated with increased stability. The Lovastatin complex demonstrated a greater number of low-energy basins for the AChE when compared to Fumitremorgin C and Hericenone J, with energy ranges of 0–5.85 kJ/mol, 0.29–6.98 kJ/mol, and 0–6.98 kJ/mol, respectively. In the BuChE-ligand complexes, the Hericenone J exhibited a higher count of low-energy basins compared to Fumitremorgin C and Lovastatin, with energy ranges of 0–5.85 kJ/mol, 0.29–6.98 kJ/mol, and 0–6.98 kJ/mol, respectively, suggesting differences in binding stability among the complexes [73–75]. The stability and interactions of AChE-ligand and BuChE-ligand complexes with fungal metabolites Fumitremorgin C, Hericenone J, and Lovastatin showed that Hericenone J is the most stable and prominent molecule for both enzymes. The AChE_Hericenone J had the longest stability period and the highest binding affinity, according to RMSD and RMSF studies. The Hericenone J also had the strongest hydrogen bond interactions, further stabilizing the AChE-ligand and BuChE-ligand complexes. In addition, the Hericenone J showed the lowest AChE binding energy (−1677.95 kcal/mol), suggesting a robust and sustained interaction. The Hericenone J's constant presence in low-energy conformations and favorable Gibbs free energy landscapes suggested it is the most promising stable AChE and BuChE inhibitor compared to Lovastatin and Fumitremorgin C.

## 5. Conclusion

The research underscores the pressing need for novel therapies for AD owing to the constraints of traditional treatments. Fungal metabolites were assessed for dual inhibition of AChE and BuChE by CADD. Of the evaluated compounds, Hericenone J exhibited the most potent and persistent interactions with both enzymes, with the highest binding affinity (−1677.95 kcal/mol) and substantial hydrogen bonding. Molecular dynamics studies validated its enhanced stability,

minimal free energy, and persistent ligand-enzyme interactions. Moreover, pharmacokinetic research demonstrated superior oral bioavailability, permeability across the blood-brain barrier, and no toxicity. These results indicate that Hericenone J is a potential multi-target lead molecule for AD treatment, necessitating more preclinical assessment.

## Supporting information

**S1 Table. The bioactive compounds of the fungal metabolites with their MeFSAT identifiers, PubChem ID, fungal metabolite name, chemical formula, SMILES, and chemical structures.**
(DOCX)

**S2 Table. Pharmacokinetics of the fungal metabolites predicted by SwissADME.**
(DOCX)

**S3 Table. Drug-likeness and bioavailability of the fungal metabolites by SwissADME.**
(DOCX)

**S4 Table. Toxicity analysis of the fungal metabolites by ProTox 3.0.**
(DOCX)

## Author contributions

**Conceptualization:** Md. Habib Ullah Masum.

**Data curation:** Md. Habib Ullah Masum, Syed Mohammad Lokman, Rehana Parvin, Md Shahidur Rahman, Erfanul Haq Chowdhury, Kazi Chamonara, Salma Chowdhury, Ahmad Abdullah Mahdeen, Mst. Mitu Khatun.

**Formal analysis:** Md. Habib Ullah Masum, Syed Mohammad Lokman, Rehana Parvin, Md Shahidur Rahman, Erfanul Haq Chowdhury, Kazi Chamonara, Salma Chowdhury, Ahmad Abdullah Mahdeen, Mst. Mitu Khatun.

**Investigation:** Md. Habib Ullah Masum.

**Methodology:** Md. Habib Ullah Masum, Syed Mohammad Lokman, Rehana Parvin, Md Shahidur Rahman, Erfanul Haq Chowdhury, Kazi Chamonara, Salma Chowdhury, Ahmad Abdullah Mahdeen.

**Resources:** Md. Habib Ullah Masum.

**Software:** Md. Habib Ullah Masum, Syed Mohammad Lokman.

**Supervision:** Md. Habib Ullah Masum.

**Validation:** Md. Habib Ullah Masum, Syed Mohammad Lokman, Rehana Parvin.

**Visualization:** Md. Habib Ullah Masum, Syed Mohammad Lokman.

**Writing – original draft:** Md. Habib Ullah Masum, Syed Mohammad Lokman, Rehana Parvin.

**Writing – review & editing:** Md. Habib Ullah Masum, Syed Mohammad Lokman.

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
