## [Decision Letter · Decision Letter 0]

PONE-D-25-16513Novel Fungal Metabolites as Dual Cholinesterase Inhibitors: A Computational Approach for Alzheimer's Disease TherapyPLOS ONE

Dear Dr. Masum,

Thank you for submitting your manuscript to PLOS ONE. After careful consideration, we feel that it has merit but does not fully meet PLOS ONE’s publication criteria as it currently stands. Therefore, we invite you to submit a revised version of the manuscript that addresses the points raised during the review process.

**ACADEMIC EDITOR: **

 The manuscript is reviewed and according to remarks after minor revision the manuscript can be accepted for publication in PLOS ONE. 

 Proteins codes must be referenced form recent research articles.One of the query of reviewer is why standards were not used for docking and simulation which might help in co-relation? if possible add standards.

We look forward to receiving your revised manuscript.

Kind regards,

Noor Jahan

Academic Editor

PLOS ONE

Journal Requirements:

Reviewers' comments:

Reviewer's Responses to Questions

**Comments to the Author**

1. Is the manuscript technically sound, and do the data support the conclusions?

Reviewer #1: Partly

Reviewer #2: Yes

2. Has the statistical analysis been performed appropriately and rigorously? 

Reviewer #1: No

Reviewer #2: N/A

3. Have the authors made all data underlying the findings in their manuscript fully available?

Reviewer #1: Yes

Reviewer #2: Yes

4. Is the manuscript presented in an intelligible fashion and written in standard English?

Reviewer #1: Yes

Reviewer #2: Yes

5. Review Comments to the Author

Reviewer #1: 1) Was the licensed version of pymol used for the study.

3) Proteins codes must be referenced form recent research articles.

3) Why standards were not used for docking and simulation which might help in co-relation?

4) In-vitro results could also aid valuable findings to the study and why these were not performed?

Reviewer #2: The article is well written and justifying the title and objectives. The Author used all necessary tools/softwares and parameters to explore the inhibitory potential of fungal metabolites on AChE and BuChE by CADD for the better treatment of Alzheimer Disease. The pictures, tables and figures are well justified.

6. PLOS authors have the option to publish the peer review history of their article (what does this mean?). If published, this will include your full peer review and any attached files.

Reviewer #1: No

Reviewer #2: No

---

## [Author Response · Author response to Decision Letter 1]

16 May 2025

ACADEMIC EDITOR:

The manuscript is reviewed and according to remarks after minor revision the manuscript can be accepted for publication in PLOS ONE.

• Proteins codes must be referenced form recent research articles.

Authors’ response: In accordance with your suggestions, we have incorporated recent research articles as references for the receptor proteins. The corresponding changes have been highlighted in yellow color.

• One of the query of reviewer is why standards were not used for docking and simulation which might help in co-relation? if possible add standards.

Authors’ response: We performed molecular docking studies between the receptor proteins and the control or standard drugs. However, for the molecular dynamics simulations, we used the apo structures of the receptor proteins as the standard, as it is a common practice to evaluate apo structures in comparison to their ligand-bound complexes in such studies. Studying the apo structure in molecular dynamics (MD) simulations is essential for understanding the intrinsic dynamics, flexibility, and stability of a protein in its unbound state. It serves as a baseline to compare with ligand-bound (holo) forms, helping to reveal conformational changes induced by ligand binding. Apo simulations can uncover transient or cryptic binding pockets that may not be visible in static crystal structures, which is valuable for drug discovery. Additionally, they provide insights into the structural integrity of the protein when not bound to a ligand and help elucidate potential allosteric mechanisms or conformational ensembles involved in its function.

Reviewer #1:

1) Was the licensed version of pymol used for the study.

Authors’ response: Many thanks for your contribution and critical observations on this paper. We believe, we have benefited much after going through this revision, and the manuscript’s quality has improved tremendously. Yes, we have used licensed version of pymol softaware.

2) Proteins codes must be referenced form recent research articles.

Authors’ response: In accordance with your suggestions, we have incorporated recent research articles as references for the receptor proteins. The corresponding changes have been highlighted in yellow color.

3) Why standards were not used for docking and simulation which might help in co-relation?

Authors’ response: We performed molecular docking studies between the receptor proteins and the control or standard drugs. However, for the molecular dynamics simulations, we used the apo structures of the receptor proteins as the standard, as it is a common practice to evaluate apo structures in comparison to their ligand-bound complexes in such studies. Studying the apo structure in molecular dynamics (MD) simulations is essential for understanding the intrinsic dynamics, flexibility, and stability of a protein in its unbound state. It serves as a baseline to compare with ligand-bound (holo) forms, helping to reveal conformational changes induced by ligand binding. Apo simulations can uncover transient or cryptic binding pockets that may not be visible in static crystal structures, which is valuable for drug discovery. Additionally, they provide insights into the structural integrity of the protein when not bound to a ligand and help elucidate potential allosteric mechanisms or conformational ensembles involved in its function.

The references are provided below

References:

1. Ibrahim, M.A.A., Abdeljawaad, K.A.A., Roshdy, E. et al. In silico drug discovery of SIRT2 inhibitors from natural source as anticancer agents. Sci Rep 13, 2146 (2023). https://doi.org/10.1038/s41598-023-28226-7

2. Akash, S., Bayıl, I., Hossain, M.S. et al. Novel computational and drug design strategies for inhibition of human papillomavirus-associated cervical cancer and DNA polymerase theta receptor by Apigenin derivatives. Sci Rep 13, 16565 (2023). https://doi.org/10.1038/s41598-023-43175-x

3. El Hassab, M.A., Eldehna, W.M., Hassan, G.S. et al. Multi-stage structure-based virtual screening approach combining 3D pharmacophore, docking and molecular dynamic simulation towards the identification of potential selective PARP-1 inhibitors. BMC Chemistry 19, 30 (2025). https://doi.org/10.1186/s13065-025-01389-2

4) In-vitro results could also aid valuable findings to the study and why these were not performed?

Authors’ response: In-vitro experiments could undoubtedly contribute valuable mechanistic insights and serve to further substantiate the findings of this study. However, due to constraints related to resources, funds and the study’s primary focus on in-vivo or field-based outcomes, such experiments were not included in the current study. Nonetheless, we acknowledge the importance of in-vitro validation and consider it a logical next step in our future research to strengthen and expand upon the current findings.

Reviewer #2: The article is well written and justifying the title and objectives. The Author used all necessary tools/softwares and parameters to explore the inhibitory potential of fungal metabolites on AChE and BuChE by CADD for the better treatment of Alzheimer Disease. The pictures, tables and figures are well justified.

Authors’ response: Many thanks for your contribution and critical observations on this paper. We believe, we have benefited much after going through this revision, and the manuscript’s quality has improved tremendously.

---

## [Editor Report · Decision Letter 1]

Novel Fungal Metabolites as Dual Cholinesterase Inhibitors: A Computational Approach for Alzheimer's Disease Therapy

PONE-D-25-16513R1

Dear Dr. Masum,

We’re pleased to inform you that your manuscript has been judged scientifically suitable for publication and will be formally accepted for publication once it meets all outstanding technical requirements.

Kind regards,

Noor Jahan

Academic Editor

PLOS ONE

Additional Editor Comments (optional):

As the necessary revisions are made so manuscript should proceed for publication.
---

## [Editor Report · Acceptance letter]

PONE-D-25-16513R1

PLOS ONE

Dear Dr. Masum,

I'm pleased to inform you that your manuscript has been deemed suitable for publication in PLOS ONE. Congratulations! Your manuscript is now being handed over to our production team.

Kind regards,

on behalf of

Prof. Dr. Noor Jahan

Academic Editor

PLOS ONE